# Cellular and molecular associations with intrinsic brain organization

Guozheng Feng[1], Jiayu Chen [1], Jing Sui [2] & Vince D. Calhoun [1]✉

Understanding how cellular and molecular architecture underpins the large-scale organization of human brain function is a central challenge in neuroscience. By integrating transcriptomic (microarray and single-nucleus RNA-sequencing), molecular imaging, and neuroimaging datasets, we observe spatial correspondences indicating that the distributions of diverse cell types, neurotransmitter systems, and mitochondrial phenotypes align with intrinsic connectivity networks (ICNs). These associations extend beyond local correspondence to reflect network-level structure: inter-ICN similarity networks derived from cellular and molecular profiles recapitulate static and dynamic patterns of functional network connectivity (FNC), mirroring canonical functional domains. Mediation analyses reveal that specific ICNs mediate the relationship between microscale cell-type architecture and domain-specific cognitive processes, while FNCs capture mediating pathways linking cell-type and neurotransmitter similarity networks to cognitive organization. Together, our findings show that the brain's functional architecture systematically aligns with cellular and molecular organization, which may constrain functional network formation and contribute to the neural basis of cognition.

Advances in the study of brain networks with resting-state functional MRI (rs-fMRI) have significantly deepened our understanding of both typical and atypical neural activity[1-6]. Among various analytic approaches, decomposition-based independent component analysis (ICA) enables the reliable extraction of intrinsic connectivity networks (ICNs), which reflect coherent patterns of spontaneous brain activity within distinct functions[7-9]. Each ICN comprises a spatial weight map paired with an associated time course (TC), capturing within-component connectivity in the blood-oxygenation-level-dependent (BOLD) signal[10]. Correlating these TCs yields between-component relationships, typically summarized as functional network connectivity (FNC)[11]. The macroscopic functional brain networks emerge from the dynamic interplay of distributed neural circuit architectures, whose region-specific implementations—distinct cellular compositions and specialized input–output motifs—necessitate tailored neuromodulatory regulation and mitochondrial bioenergetics to sustain network-specific information processing and integration[12-21]. Despite growing

evidence for the relevance of these cellular and molecular factors, their specific contributions to the organization of functional networks remain largely uncharacterized.

The growing availability of high-resolution cellular and molecular atlases of the human brain—such as microarray-based transcriptional data from the Allen Human Brain Atlas (AHBA)[22], single-nucleus RNA sequencing (snRNA-seq) datasets from discrete brain regions[21,23,24], and positron emission tomography (PET) imaging of diverse neurotransmitter systems[25]—has created new opportunities to bridge microscale biology with macroscale brain organization. Previous work established spatial associations between the distribution of interneuron-linked genes and regional differences in fMRI signal variability[26,27]. The low-dimensional manifold organizing fMRI activity shows opposing associations with regional densities of facilitatory versus inhibitory neuromodulatory receptors[28]. In parallel, specific mitochondrial phenotypes selectively align with distinct fMRI signal metrics (e.g., maximum BOLD, regional homogeneity, entropy)[21].

[1]Tri-Institutional Center for Translational Research in Neuroimaging and Data Science (TReNDS), Georgia State University, Georgia Institute of Technology, and Emory University, Atlanta, GA, USA. [2]State Key Laboratory of Cognitive Neuroscience and Learning & IDG/McGovern Institute for Brain Research, Beijing Normal University, Beijing, China. ✉e-mail: vcalhoun@gatech.edu

Beyond spatial alignment, molecular similarity further supports the organization of functional networks: functionally coupled regions tend to share gene-expression signatures[29–31], and distinct cell-type distributions differentially align with the cortex's principal unimodal-to-transmodal functional gradient[32]. Concordantly, neurotransmitter receptor and transporter similarity across regions correlates with functional connectivity, with brain areas exhibiting similar receptor profiles showing stronger co-activation[25]. Although most prior studies have focused on individual neurobiological scales, observed correlations among neurobiological scales[25,33] suggest that multi-scale factors are likely to synergistically contribute to shape functional network phenotypes. One illustrative case is that cholinergic neurotransmission primarily modulates neuronal excitability, regulates presynaptic transmitter release, and orchestrates the firing of neuronal populations[34–36]. Additionally, developmental patterns of cortical thickness are better captured by models that integrate the spatial distributions of several systems (e.g., cell-type composition, neurotransmitter receptor/transporter densities), with neurotransmitter markers showing relatively greater scale-level relative importance[37]. Together, integrating diverse cellular and molecular atlases provides a multi-scale perspective for revealing the biological foundations of intrinsic functional networks.

The human brain supports a wide range of cognitive and behavioral functions through large-scale neural circuits shaped by its underlying cellular and molecular architecture[17]. At the macroscale, ICNs, as the core organizational units of functional architecture, along with their static and dynamic FNCs, demonstrate heterogeneous contributions to diverse cognitive abilities[38,39]. At the microscale, specific inhibitory neuron subtypes are selectively preserved in individuals who maintain high cognitive performance in late life, implicating cell-type-specific influences on cognitive resilience[40]. The spatial distribution of neurotransmitter receptors differentiates intrinsic and extrinsic cognitive modes[25]. Animal studies further underscore the role of molecular processes in cognition and behavior: mitochondrial function has been shown to impact both cognitive capacity[41,42] and behavioral regulation[43,44], while distinct neurotransmitter receptor systems have been associated with specific cognitive domains[45]. Despite these converging findings, the mechanisms through which ICNs and FNCs mediate the relationship between microscale biological substrates and cognitive and behavioral abilities remain poorly understood. Uncovering this linkage is essential for a deeper understanding of how cellular and molecular systems scaffold the brain's functional architecture and enable complex cognition.

Here, motivated by the above evidence, we aim to probe the cellular and molecular bases of the spatial and connectional architecture of ICA-derived ICNs from a multi-scale perspective. For this purpose, we integrate spatial maps of cell types, neuromodulatory receptor/transporter systems, and mitochondrial phenotypes. We hypothesize that ICN spatial maps colocalize with these microscale architectures—such that networks are differentially enriched for several biological features that support network-specific information processing and integration. We further hypothesize that between-region similarity network derived from these cellular and molecular features is associated with FNC, consistent with an organizing role for these similarity networks. Moreover, multimodal combinations of neurobiological features explain more ICN/FNC variance than any single system, consistent with potential synergistic effects among distinct microscale processes. Finally, we posit that ICNs and FNCs provide the macroscale substrates that statistically link microscale neurobiology to cognitive/behavioral maps.

## Results

We analyzed 53 reproducible ICN spatial maps and their corresponding FNC (Supplementary Fig. S1 and Fig. 1a) derived from the Neuromark_fMRI_1.0 template[7]. The NeuroMark template was based on

components that replicated across two large-scale datasets: the Genomics Superstruct Project (GSP; $n = 1005$; mean age = 21.54 ± 2.96 years; 411 males)[46] and the Human Connectome Project (HCP; $n = 823$; mean age = 28.79 ± 3.68 years; 356 males)[47]. To investigate the cellular and molecular architecture underlying ICNs, we systematically examined the spatial and connectional correspondence between these functional networks and a diverse sets of cell-type[23], neurotransmitter[25], and mitochondrial[21] maps (Fig. 1b). Extending this analysis, we incorporated 123 cognitive and behavioral probabilistic maps from Neurosynth[48] to assess whether functional networks (ICNs and FNCs) mediate the relationship between microscale biological substrates and spatial and connectional variations in cognitive and behavioral functions.

### Cell-type associations of ICNs

To investigate the cellular architecture underlying ICNs, we related postmortem gene expression–based estimates of cell-type distribution to the spatial maps of 44 cortical ICNs. Bulk tissue transcriptomic data from AHBA[22] were processed using an established harmonization pipeline[49–51]. Cell-type–specific transcriptional signatures were derived from snRNA-seq of eight cortical areas[23], enabling the imputation of 24 cell-type abundances across AHBA samples. These included nine excitatory glutamatergic neurons (e.g., L2/3 IT, L4IT, L6IT), nine inhibitory GABAergic interneurons (e.g., Chandelier, PVALB, SST), and six non-neuronal cell types (e.g., Astro, Endo, OPC), each characterized by distinct laminar locations, developmental origins, morphology, and projection targets[23,52]. Voxel-level spatial distributions of cell-type abundance were correlated with ICN spatial maps using Spearman's $r$ correlation. Statistical significance was determined using a permutation-based two-sided Moran test[53] (1000 iterations), with false discovery rate (FDR) correction applied across 24 × 44 comparisons.

Seven ICNs exhibited at least one significant association with the imputed abundance of specific cell types (Fig. 2a). These associations included both positive and negative correlations, indicating that cell-type gradients may support the spatial organization of ICNs. IC14 (precentral gyrus network) in the sensorimotor network (SM) domain was positively associated with L5ET ($r (107) = 0.48$, $q < 0.001$). IC25 (middle temporal gyrus network) in the visual network (VIS) domain was positively associated with L2/3IT ($r (140) = 0.30$, $q < 0.001$). In the cognitive-control network (CC) domain, IC27 (insula network) was significantly associated with L6ITCar3 ($r (127) = 0.58$, $q < 0.001$), and IC28 (superior medial frontal gyrus network) was negatively associated with L4IT ($r (123) = −0.58$, $q < 0.001$), IC31 (middle frontal gyrus network) was positively associated with LAMP5 ($r (110) = 0.49$, $q < 0.001$), IC40 (inferior frontal gyrus network) was negatively associated with L5/6NP ($r (88) = −0.40$, $q < 0.001$). IC45 (anterior cingulate cortex network) in the default-mode network (DM) domain was positively associated with SST ($r (139) = 0.32$, $q < 0.001$) and negatively associated with OPC ($r (139) = −0.25$, $q < 0.001$). The spatial distributions for all 24 cell types are presented in Fig. 2b. Collectively, these findings indicate that particular ICNs show specific spatial correspondences with distinct cell types.

### Neurotransmitter and mitochondrial associations of ICNs

To examine the molecular architecture underlying ICNs, we assessed the spatial alignment between ICN maps and neurotransmitter systems using PET-derived estimates of receptor and transporter densities. We compiled 19 neurotransmitter maps encompassing nine neurotransmitter systems, aggregated from PET imaging data across more than 1200 healthy individuals[25]. These maps included serotonergic (5-HT$_{1A}$, 5-HT$_{1B}$, 5-HT$_{2A}$, 5-HT$_4$, 5-HT$_6$, 5-HTT), dopaminergic (D$_1$, D$_2$, DAT), noradrenergic (NET), histaminergic (H$_3$), cholinergic ($\alpha_4\beta_2$, M$_1$, VAChT), cannabinoid (CB$_1$), opioid (MOR), glutamatergic (NMDA, mGluR$_5$), and GABAergic (GABA$_A$) systems.

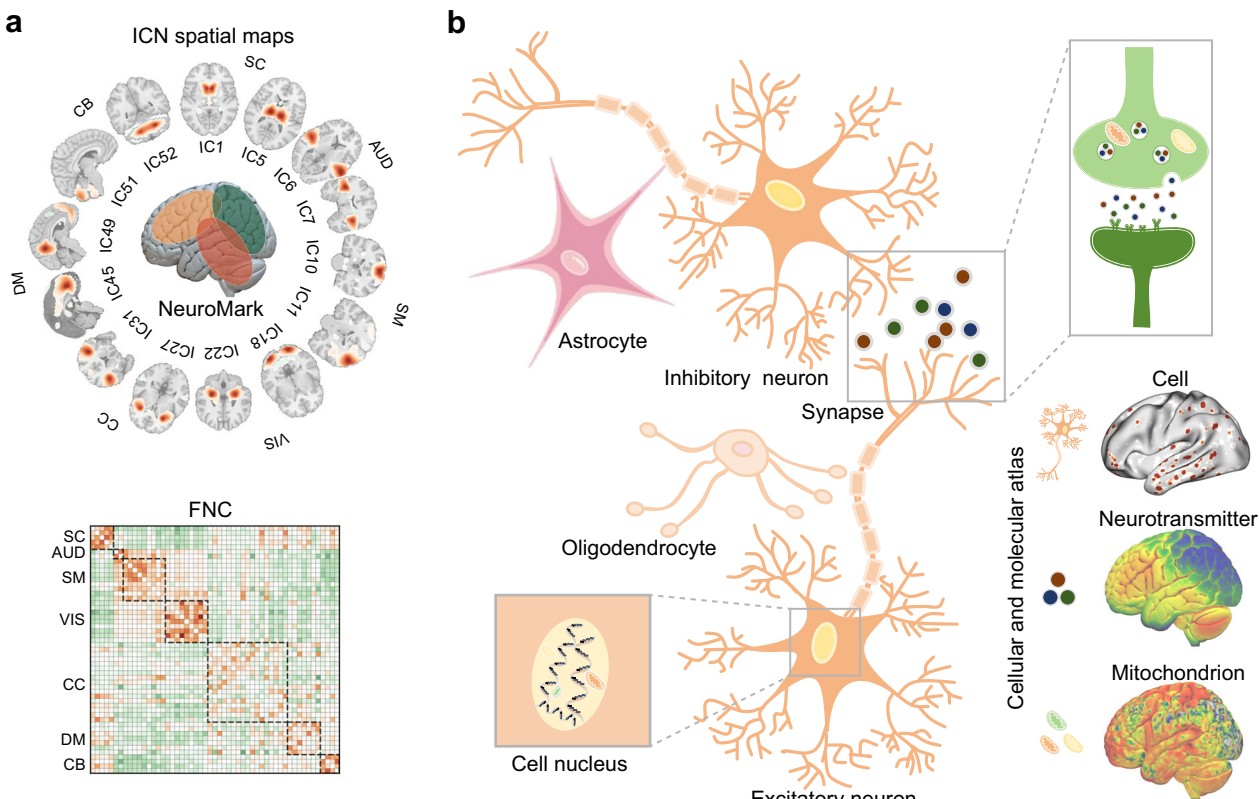

**Fig. 1 | ICNs and cellular and molecular processes. a** Fifty-three robust and replicable ICNs were identified and organized into seven canonical functional domains using Neuromark_fMRI_1.0[7], derived from rs-fMRI data of 1,828 young adults across the GSP and HCP cohorts. Each ICN comprises brain voxels that co-vary over time, capturing intra-network coherence. Functional connectivity between ICNs (FNC) was derived by quantifying temporal correlations between ICN TCs, reflecting inter-network coupling. **b** A diverse set of cellular and molecular maps derived from transcriptomic data (microarray and snRNA-seq) and molecular imaging data (PET), illustrates regional variations in cellular composition, mitochondrial metabolism, and neurotransmitter signaling, providing a multiscale context for interpreting ICN organization.

For each ICN, we constructed multilinear regression models using the full set of receptor and transporter densities as predictors. Statistical significance was assessed using a one-sided Moran test (1000 permutations) to account for spatial autocorrelation[53], with FDR correction applied across 53 comparisons. Sixteen ICNs exhibited significant model fits (adjusted $r^2$: 0.43–0.86, $q < 0.05$, Fig. 3a), indicating that distinct combinations of neurotransmitter systems spatially align with specific ICN architecture; effects were densest in subcortical network (SC) and auditory network (AUD) domains. To determine the relative contribution of individual neurotransmitter systems, we applied dominance analysis, which quantifies the proportion of variance in each ICN map attributable to each receptor/transporter in the model[54]. Dominance scores were normalized by the model's total adjusted $r^2$, enabling comparison across ICNs (Fig. 3b). Statistical significance of neurotransmitter-specific dominance was assessed using one-sided Moran test (1000 permutations) to account for spatial autocorrelation[53], FDR correction applied across 19 × 53 comparisons. Variance contributions of neurotransmitter receptors were heterogeneous across ICNs. For example, in CC domain, the $\alpha_4\beta_2$ receptor made a prominent contribution to IC31 (middle frontal gyrus network, contribution = 26%, $q < 0.001$), while $CB_1$ receptor made a prominent contribution to IC37 (hippocampus network, contribution = 19%, $q < 0.001$). In DM domain, 5-HT$_{1B}$ receptor showed a significant influence on IC48 (precuneus network, contribution = 25%, $q < 0.001$).

In parallel, we examined the associations between mitochondrial phenotypes and ICNs. Six mitochondrial maps (complex I [CI], complex II [CII], complex IV [CIV], mitochondrial density [MitoD], tissue respiratory capacity [TRC], and mitochondrial respiratory capacity [MRC])[21] were regressed onto ICN spatial maps, with FDR correction applied across 53 comparisons for adjusted $r^2$ and 6 × 53 comparisons for dominance. We observed significant model fits for IC2 (adjusted $r^2 = 0.25$, $p < 0.001$) and IC4 (caudate network, adjusted $r^2 = 0.24$, $p < 0.001$), both located in SC domain (Supplementary Fig. S2a). Within these models, MitoD, and CIV each made significant relative contributions (Supplementary Fig. S2b). These findings suggest that mitochondrial energy metabolism may contribute to the functional architecture of specific ICNs in SC domain.

To investigate the synergy and relative contributions of distinct neurobiological scales to ICNs, we examined associations between multi-scale markers (cell types [21, after collinearity attenuation, Supplementary Fig. S3; see "Methods"], neurotransmitters [19], and mitochondria [6]) and ICNs. Fourteen ICNs exhibited significant model fits (adjusted $r^2$: 0.62–0.99, $q < 0.05$; Supplementary Fig. S4a). In 13 of these 14 ICNs, multi-scale models outperformed single-scale models ($q < 0.05$; Supplementary Fig. S4b), underscoring the synergistic value of integrating distinct neurobiological scales. We further observed a consistent trend of synergistic contributions from markers across scales ($p < 0.05$, Supplementary Fig. S4a). Among these, neurotransmitter systems consistently accounted for the largest overall relative contributions across scales (Supplementary Fig. S4b).

## Cellular and molecular similarity networks relate to functional connectivity

Having established that ICNs exhibit distinct spatial associations with cellular and molecular features, we next asked whether ICNs that are similarly composed at the cellular and molecular levels also exhibit

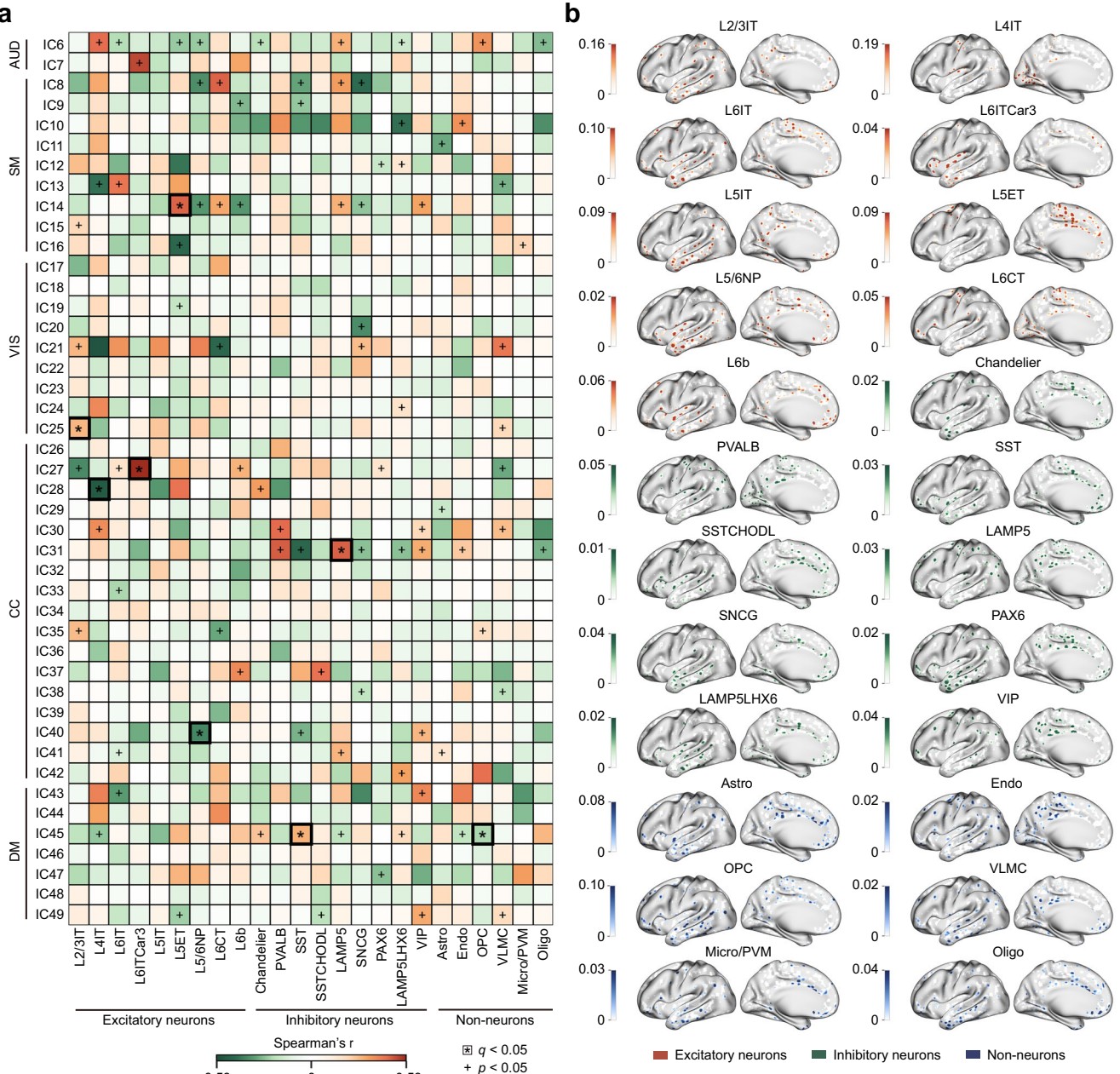

**Fig. 2 | Cell-type associations of ICNs.** Imputed spatial distributions of 24 cortical cell types were derived by integrating bulk tissue transcriptomic data from the AHBA[22] with snRNA-seq from eight cortical regions[23,52]. Voxel-wise abundance maps were correlated with the spatial patterns of 44 cortical ICNs using Spearman's correlation. Statistical significance was determined using a two-sided Moran test[53] (1000 iterations), with FDR correction applied across 24 × 44 comparisons. **a** Eight significant cell-type–ICN associations (positive and negative). **b** Cortical distribution maps of the 24 cell types. AUD auditory network, SM sensorimotor network, VIS visual network, CC cognitive-control network, DM default-mode network. Source data are provided as a Source Data file.

stronger functional interactions. To this end, we computed similarity networks for cell-type, mitochondrial, neurotransmitter, and combined molecular fingerprints by correlating the mean spatial profiles of each ICN pair using Spearman's correlation (Fig. 4a). Across all modalities, cellular and molecular similarities were significantly greater among ICNs within the same functional domain than between different domains, even after regressing out Euclidean distance and spatial overlap (Dice coefficient) from the similarity matrices ($p < 0.001$, ranksum test; $N_{within} = 230$ edges, $N_{between} = 716$ edges for cell-type; $N_{within} = 246$ edges, $N_{between} = 1132$ edges for others, Fig. 4b). We then assessed the relationship between cellular/molecular similarity and functional connectivity, as indexed by the group-level FNC from the HCP dataset (Fig. 4c). We observed that cell-type similarity

(Spearman's $r$ (946) = 0.19, $p < 0.001$), neurotransmitter similarity (Spearman's $r$ (1378) = 0.27, $p < 0.001$), and the combined similarity (Spearman's $r$ (1378) = 0.46, $p < 0.001$) networks were all significantly correlated with FNC strength (Fig. 4d). At the regional level, correlations were predominantly positive, with the strongest associations observed in nodes located within the SM and VIS domains (Fig. 4e). To further investigate the dynamic aspects of this relationship, we identified five discrete states of dynamic FNC (Supplementary Fig. S5a) and examined their correspondence with cellular and molecular similarity networks. Results consistently demonstrated robust associations between FNC states and similarity networks, both at the global (Supplementary Fig. S5b–f) and regional (Supplementary Fig. S6a–e) levels. Together, these results suggest that ICNs with

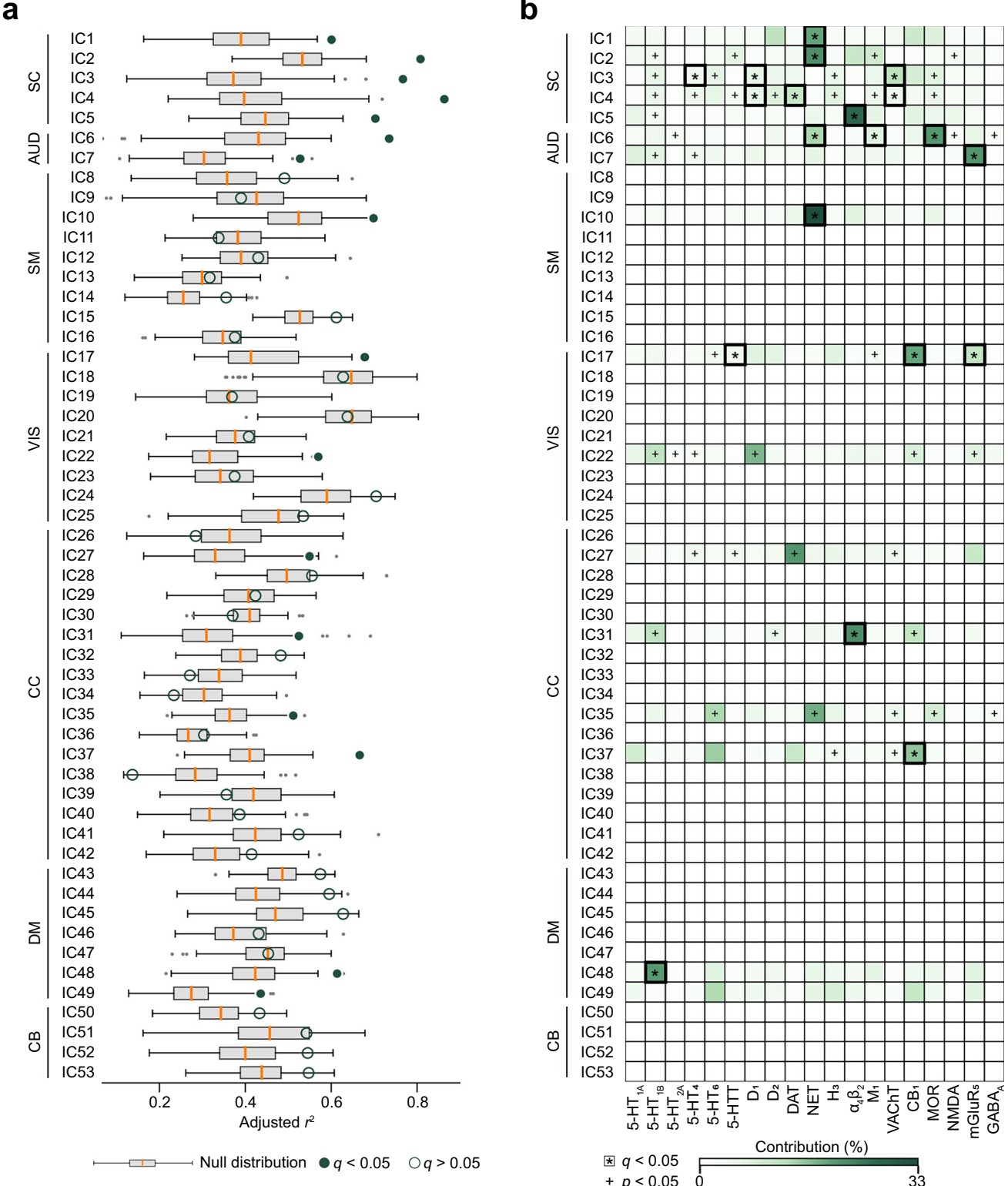

**Fig. 3 | Neurotransmitter associations of ICNs.** A multilinear regression framework was used to evaluate the spatial correspondence between 53 ICNs and the distributions of 19 neurotransmitter receptors and transporters, as estimated by PET imaging[25]. Dominance analysis was performed to quantify the unique variance in each ICN map attributable to individual receptor/transporter densities[54]. Statistical significance was assessed using one-sided Moran test (1000 permutations) to account for spatial autocorrelation[53], with FDR correction applied across 53 adjusted $r^2$ comparisons and 19 × 53 dominance comparisons. **a** Sixteen ICNs demonstrated significant model fits (0.43 ≤ adjusted $r^2$ ≤ 0.86). Box plots show the median and interquartile range (IQR; 25–75%), with whiskers indicating 1.5× IQR from the first or third quartile. **b** Dominance analysis showed that variability across ICNs was selectively explained by distinct receptor/transporter profiles. SC subcortical network, AUD auditory network, SM sensorimotor network, VIS visual network, CC cognitive-control network, DM default-mode network, CB cerebellar network. Source data are provided as a Source Data file.

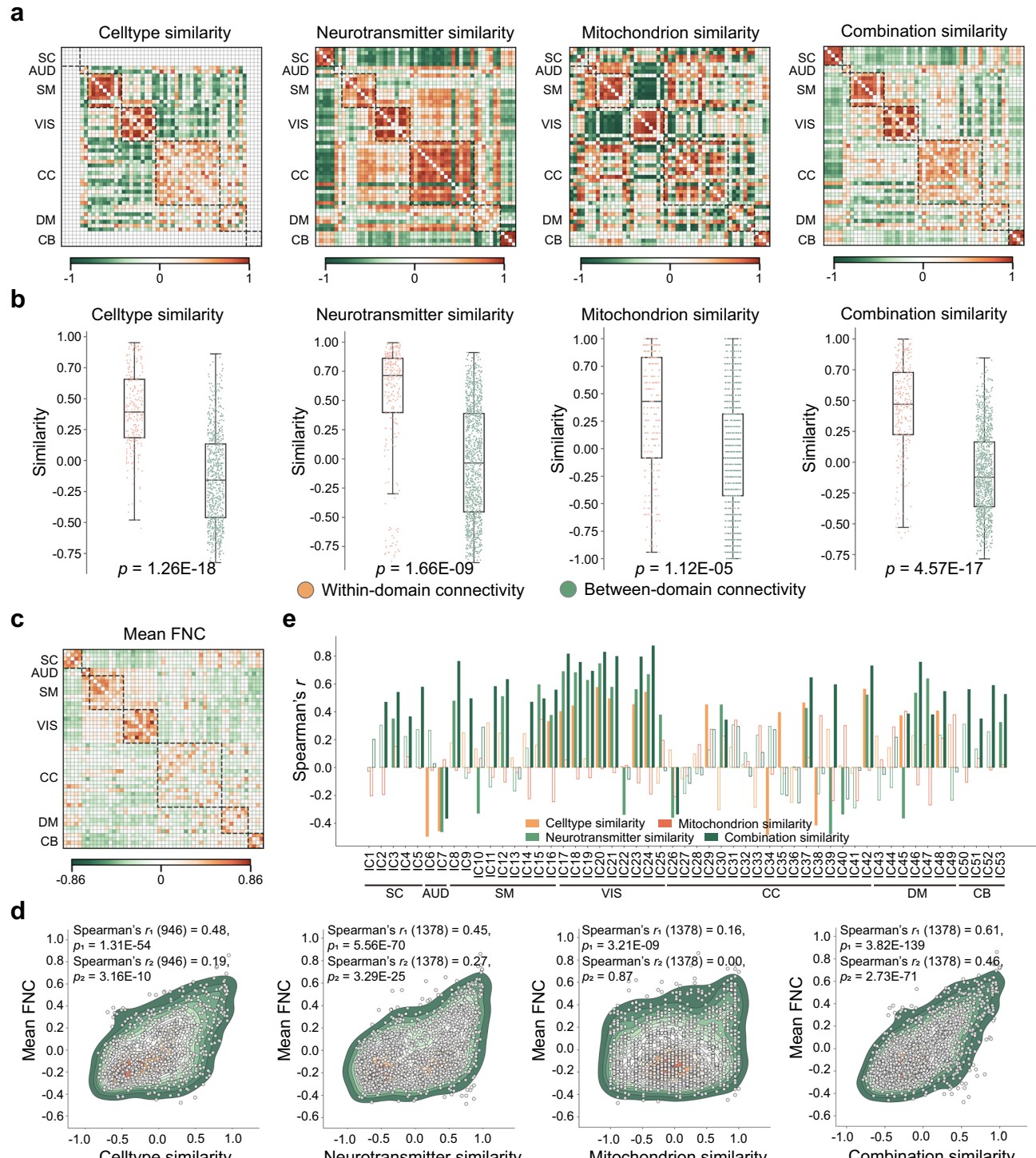

**Fig. 4 | Cellular and molecular similarity networks relate to functional connectivity.** **a** Similarity networks were constructed for cell-type composition, mitochondrial metabolism, neurotransmitter receptor density, and an integrated molecular profile by computing pairwise Spearman correlations between the spatial fingerprints of all ICN pairs. **b** Within-domain versus between-domain similarity was compared across these networks using a two-sided rank-sum test ($N_{within}$ = 230 edges, $N_{between}$ = 716 edges for cell-type; $N_{within}$ = 246 edges, $N_{between}$ = 1132 edges for others), with confounding effects of spatial proximity (Euclidean distance and Dice similarity) regressed out. Box plots show the median and interquartile range (IQR; 25–75%), with whiskers indicating 1.5× IQR from the first or third quartile.

**c** Mean FNC. Global (**d**) and regional (**e**) associations between FNC and cellular/ molecular similarity networks were evaluated using two-sided partial Spearman's correlation, controlling for spatial proximity. In **e**, filled bars indicate statistically significant associations ($q < 0.05$), whereas unfilled bars denote non-significant results. Note: $r_1$ and $p_1$ denote the correlation coefficient and p-value without controlling for spatial proximity; $r_2$ and $p_2$ denote those after controlling for spatial proximity. SC subcortical network, AUD auditory network, SM sensorimotor network, VIS visual network, CC cognitive-control network, DM default-mode network, CB cerebellar network. Source data are provided as a Source Data file.

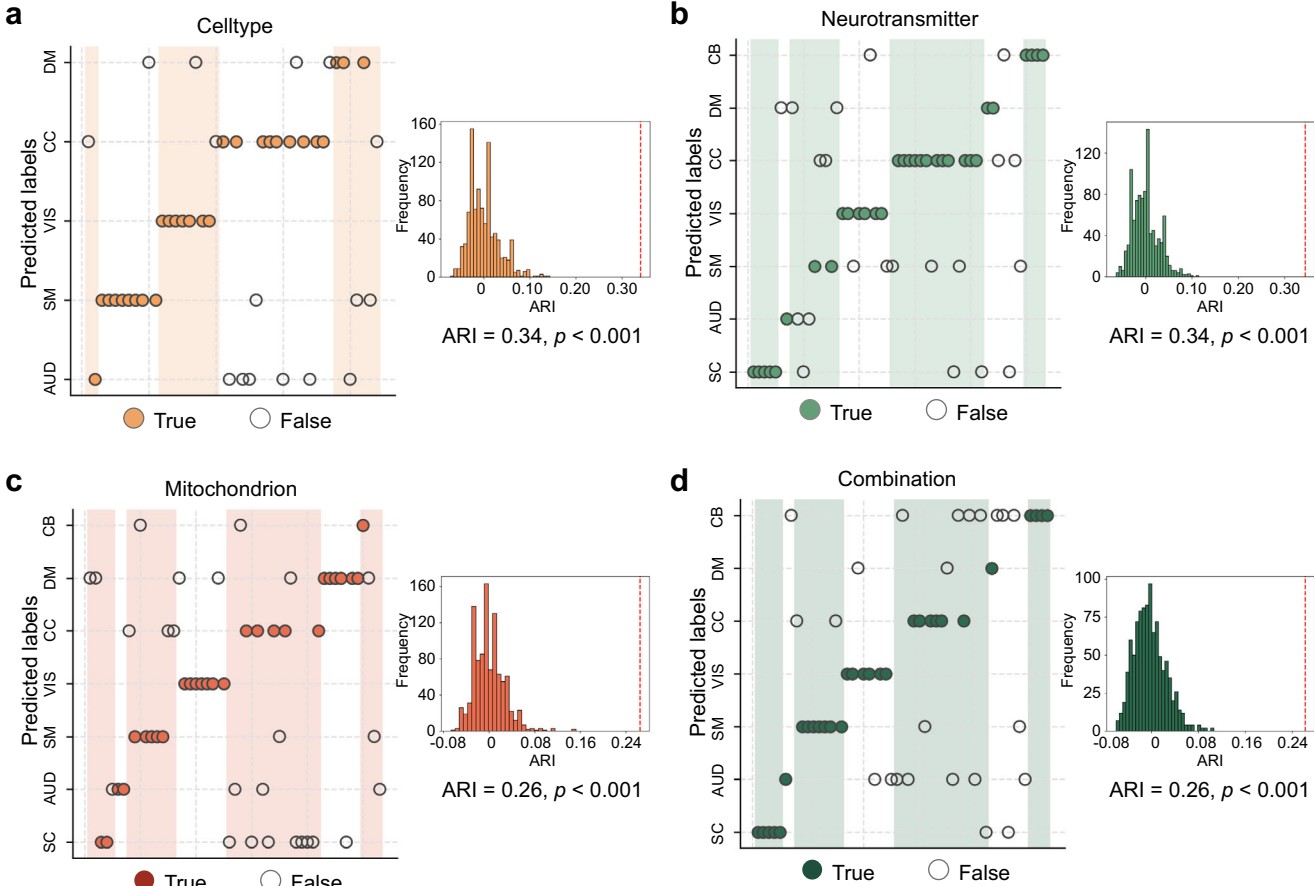

**Fig. 5 | Structural associations between ICNs and cellular/molecular architecture.** Based on constructed cellular and molecular similarity networks controlling for spatial proximity across ICNs, we applied a diffusion embedding method[55] to project these similarity networks into a shared three-dimensional space. *k*-means clustering was then performed on the embedded features, with the number of clusters determined by the number of established functional domains: five clusters for the cell-type similarity network (**a**), and seven for the neurotransmitter (**b**), mitochondrial (**c**), and combined (**d**) networks. The labels derived from this unsupervised clustering were compared to the true functional domain labels of the ICNs. The adjusted rand index (ARI) was used to quantify the accuracy of the clustering, and 1000 label permutations were performed to assess the statistical significance of the results using a one-sided permutation test. SC subcortical network, AUD auditory network, SM sensorimotor network, VIS visual network, CC cognitive-control network, DM default-mode network, CB cerebellar network. Source data are provided as a Source Data file.

similar cellular and molecular compositions are more likely to be functionally coupled, independent of spatial proximity, supporting the idea that inter-regional communication is constrained by a common microscale architecture.

## Identify cellular and molecular structural associations with functional domain

To test whether cellular and molecular profiles reflect the functional domain architecture of ICNs, we constructed similarity networks based on cell-type, neurotransmitter, mitochondrion and combination fingerprints, while regressing out the effects of Euclidean distance and Dice similarity from the network matrices. We reduced network connectivity features using diffusion embedding[55], and projected each similarity network into a three-dimensional space (cumulative variance explained: 57% for cell type, 51% for neurotransmitter, 54% for mitochondrion, and 50% for the combined profile). Component-wise explained variance is reported in Supplementary Fig. S7. Comparisons of dimensionality reduction methods are shown in Supplementary Fig. S8. We then applied *k*-means to cluster features of cellular/molecular similarity network. The ground-truth labels were derived from the functional domain assignments of ICNs, and clustering performance was quantified using the adjusted rand index (ARI), with significance assessed via a null generated from 1000 label

permutations. Clustering based on cell-type (ARI = 0.34, *p* < 0.001, 5 clusters; Fig. 5a), neurotransmitter (ARI = 0.34, *p* < 0.001, 7 clusters; Fig. 5b), mitochondrion (ARI = 0.26, *p* < 0.001, 7 clusters; Fig. 5c), and combined profiles (ARI = 0.26, *p* < 0.001, 7 clusters; Fig. 5d) all showed significant alignment with functional domain structures. These findings indicate that the spatial organization of functional domains is supported by coherent cellular and molecular architectures.

## ICNs mediate spatial associations between cellular/molecular architecture and cognitive function

To examine whether and how ICNs mediate the relationship between microscale cellular and molecular organization and macroscale cognitive architecture, we leveraged probabilistic meta-analytic activation maps from Neurosynth (https://github.com/neurosynth/neurosynth)[48]. These maps encode the likelihood that a given cognitive term is associated with activation at each voxel, based on over 15,000 published fMRI studies. A total of 123 cognitive and behavioral terms were selected based on their functional annotations in the Cognitive Atlas[56]. These term-wise maps showed significant spatial correspondence with specific ICNs (Supplementary Fig. S9). For cellular/molecular maps and ICNs that showed significant spatial correspondence in the primary analyzes, we then conducted post-hoc mediation analyzes using the PROCESS toolbox in R. In each model, the

cellular/molecular map served as the predictor, the cognitive probability map as the outcome, and the ICN as the mediator. FDR correction was applied across 123 × 30 comparisons.

These analyzes revealed significant mediation pathways, predominantly involving 7 ICNs, 10 cellular/molecular maps, and 123 cognitive and behavior functions. Among cortical ICNs (Fig. 6), the insula network (IC27; L6ITCar3-coupled) showed the broadest and strongest mediation spectrum (mediation effect $\beta$: 0.21–0.57), spanning language (speech perception/production, naming), sensory integration (multisensory, categorization), executive control (cognitive control, inhibition, decision making), and emotion-related processes (reward anticipation, anxiety, fear). The precentral gyrus network (IC14; L5ET-coupled) exhibited weaker mediation for executive/memory functions but stronger effects on perceptual attention and movement/motor control (mediation effect $\beta$: 0.10–0.23). The superior medial frontal gyrus network (IC28; L4IT-coupled) showed modest effects focused on planning and attention, with additional links to movement (mediation effect $\beta$: 0.11–0.19). Finally, the anterior cingulate cortex network (IC45; SST-coupled) preferentially mediated affective and motivational processes, including fear, anxiety, mood regulation, and reinforcement learning (mediation effect $\beta$: 0.13–0.21). By comparison, neurotransmitter and mitochondrial markers primarily mediated subcortical ICN–cognition associations (Source Data Table S1). Overall, these spatial association–mediation results indicate that specific cellular/molecular markers interface with functional networks to provide indirect links to cognitive and behavioral phenotypes.

## FNCs mediate the connectional associations between cellular and molecular architecture and cognitive function

To further investigate whether inter-ICN functional connectivity (static and dynamic FNCs) mediates the relationship between cellular/molecular similarity and cognitive similarity, we first averaged the cognitive probabilistic maps across the 53 ICNs to obtain ICN-level cognitive features. We then constructed cognitive similarity networks by computing Spearman's correlation across ICN pairs. Partial Spearman's correlations were performed to assess associations between cellular/molecular similarity networks and the cognitive similarity network, as well as between static and dynamic FNC states and the cognitive similarity network. To account for spatial confounds, Euclidean distance and Dice similarity were included as covariates in all correlation and mediation analyzes. We found that the cell-type, neurotransmitter, and combined similarity networks were significantly associated with the cognitive similarity network ($p < 0.001$, Supplementary Fig. S10). Both static FNC and dynamic FNC states showed significant associations with the cognitive similarity network ($p < 0.001$, Supplementary Fig. S11). Mediation analyzes at the network level treated the cellular or molecular similarity network as the predictor, the cognitive similarity network as the outcome, and the FNC (static or dynamic) as the mediator. The results revealed that static FNC partially mediated the relationship between the cell-type similarity network and the cognitive similarity network (path $ab$: $p = 0.002$, $\beta = 0.04$, 95% CI = [0.01, 0.06]; path $c'$: $p < 0.001$, $\beta = 0.31$, 95% CI = [0.26, 0.35]; path $c$: $p < 0.001$, $\beta = 0.35$, 95% CI = [0.30, 0.40], Fig. 7a). Static FNC also mediated the link between the neurotransmitter similarity network and cognitive similarity network (path $ab$: $p < 0.001$, $\beta = 0.05$, 95% CI = [0.03, 0.06]; path $c'$: $p < 0.001$, $\beta = 0.21$, 95% CI = [0.16, 0.25]; path $c$: $p < 0.001$, $\beta = 0.25$, 95% CI = [0.21, 0.29], Fig. 7b), as well as between the combination similarity network and the cognitive similarity network (path $ab$: $p < 0.001$, $\beta = 0.08$, 95% CI = [0.07, 0.10]; path $c'$: $p < 0.001$, $\beta = 0.36$, 95% CI = [0.32, 0.41]; path $c$: $p < 0.001$, $\beta = 0.45$, 95% CI = [0.40, 0.49], Fig. 7d). No significant mediation was detected for the mitochondrial similarity network (Fig. 7c). Mediation by dynamic FNC states was also significant as shown in Supplementary Fig. S12. Specifically, states 2–5 mediated the relationship between cell-type and cognitive similarity;

states 3–5 mediated the neurotransmitter–cognition link; states 1, 3–5 mediated the mitochondrion–cognition link. All states significantly mediated the association between combination–cognition link. Together, these findings demonstrate that static and dynamic FNCs provide indirect pathways linking cellular/molecular architecture to cognitive organization, with dynamic FNC offering additional state-specific routes.

## Discussion

By integrating transcriptomic, molecular, and neuroimaging data, we observed spatial associations between the brain's large-scale functional architecture and its underlying cellular and molecular organization. Spatial profiles of diverse cell types, neurotransmitter systems, and mitochondrial markers showed selective alignment with ICNs, providing biologically grounded support for functional segregation. These relationships extended beyond local overlap: similarity networks constructed from molecular and cellular features approximated patterns of functional connectivity and captured core ICN functional domains. Importantly, ICNs statistically served as intermediaries in the associations between molecular architecture and cognitive processes, positioning them as critical intermediaries linking microscale biology to behavior. Together, these findings suggest that cellular and molecular organization may impose constraints on the formation and function of large-scale brain networks.

rs-fMRI investigations of brain networks have substantially expanded the frontiers of cognitive neuroscience and psychiatric research. Among data-driven approaches, ICA leverages the hidden spatiotemporal structure of the BOLD signal to separate meaningful neural signals from noise, while preserving subject-level variability[57] and yielding improved signal-to-noise ratios[7]. Critically, unlike parcellation-based approaches, ICA enables the delineation of distinct patterns of functional activity, thereby yielding ICN maps[58]. Each ICN comprises brain voxels that co-vary over time, capturing within-component connectivity. Functional connectivity between ICNs (FNC) can subsequently be derived by quantifying temporal correlations across the ICA time series. To facilitate cross-study reproducibility, methodological advances have led to the fully automated spatially constrained ICA framework (NeuroMark) as a template based on 1828 young adult participants in the GSP and HCP datasets[7]. This approach has identified 53 reliable ICNs, which have since served as functional templates in studies of lifespan development[8], individual differences in cognition[38,59], and a range of neuropsychiatric disorders[60,61]. These large-scale ICNs are thought to arise from the coordinated activity of underlying neural circuits, yet the extent to which they are shaped by microscale cellular and molecular substrates remains poorly understood. Elucidating the cellular and molecular architecture underlying ICNs and their connectivity patterns (FNCs) could offer a foundational neurobiological annotation of functional network organization. Such insights are crucial for bridging levels of analysis from genes and cells to circuits and cognition.

The macroscale functional network organization of the brain is increasingly understood to reflect underlying microscale cellular and molecular architecture[18,26,32]. To explore this hypothesis, we systematically integrated multiple cellular and molecular datasets and quantified their spatial associations with ICNs. We observed spatial coupling between the distributions of select cell types and ICNs, consistent with shared structural and computational motifs. For example, L5ET neurons constitute the cortex's principal corticofugal output, routing columnar computations to thalamus, brainstem, and spinal targets[62,63]; their alignment with the precentral gyrus network (IC14) supports involvement in action control and motor execution circuits[64]. By contrast, L6ITCar3 neurons possess extensive intracortical axon projections that enable cross-areal integration[63,65], providing a cellular substrate for the insula network's (IC27) broad role as a switching hub across externally oriented and internally oriented systems[66]. SST

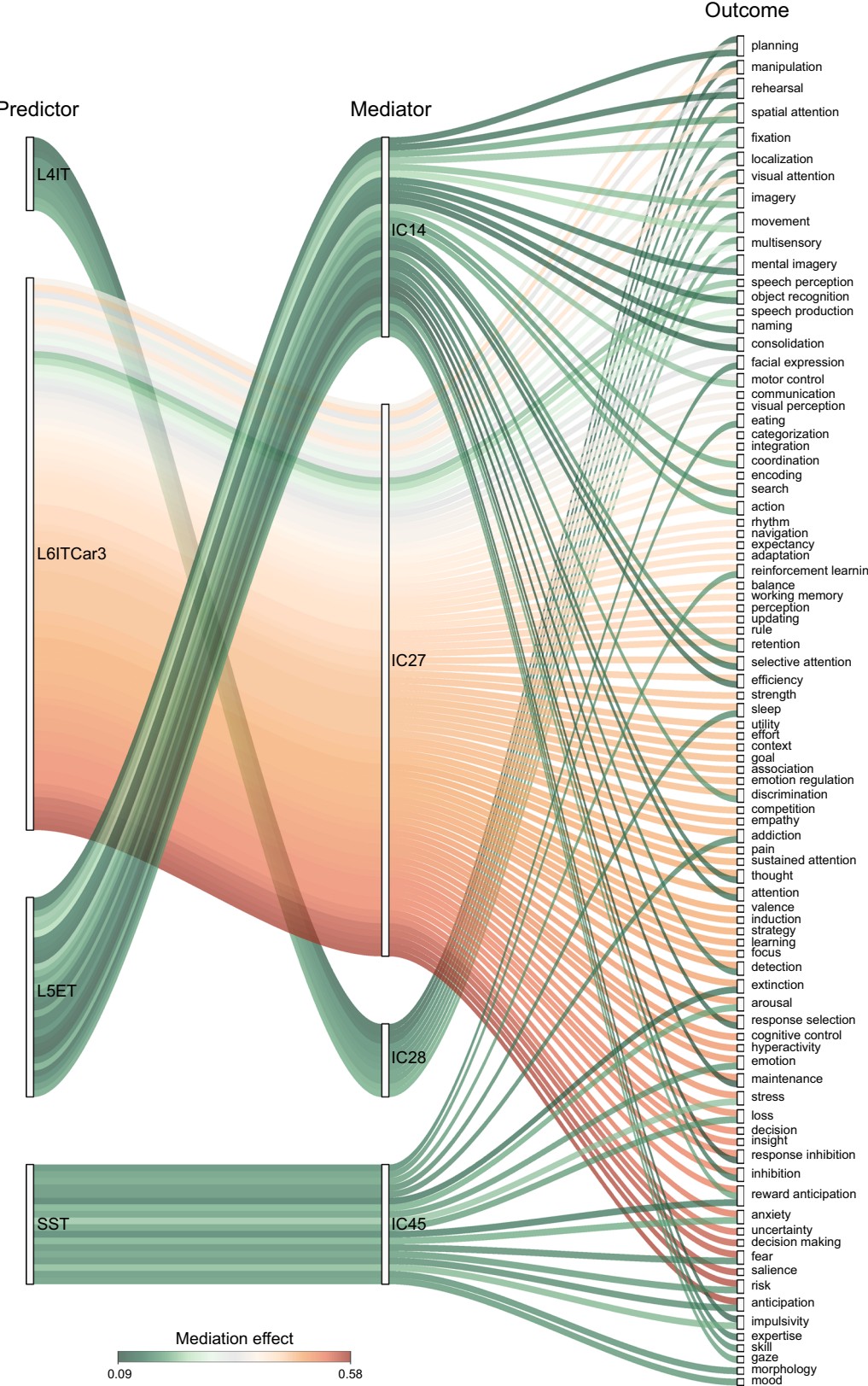

**Fig. 6 | ICNs mediate the spatial associations between cellular and molecular architecture and cognitive function.** We used 123 probabilistic cognitive activation maps derived from meta-analyses in Neurosynth[48]. For cellular/molecular maps and ICNs that showed significant spatial correspondence in the primary analyses, mediation analyses were conducted using the PROCESS toolbox in R, with cellular/molecular maps as predictors, ICNs as mediators, and cognitive maps as outcomes. Edges in the diagram represent significant mediation pathways ($q < 0.05$), with color indicating the magnitude of the mediation effect. L4IT layer 4 intratelencephalic neurons, L6ITCar3 layer 6 intratelencephalic Car3-expressing neurons, L5ET layer 5 extratelencephalic neurons, SST somatostatin interneurons. Source data are provided as a Source Data file.

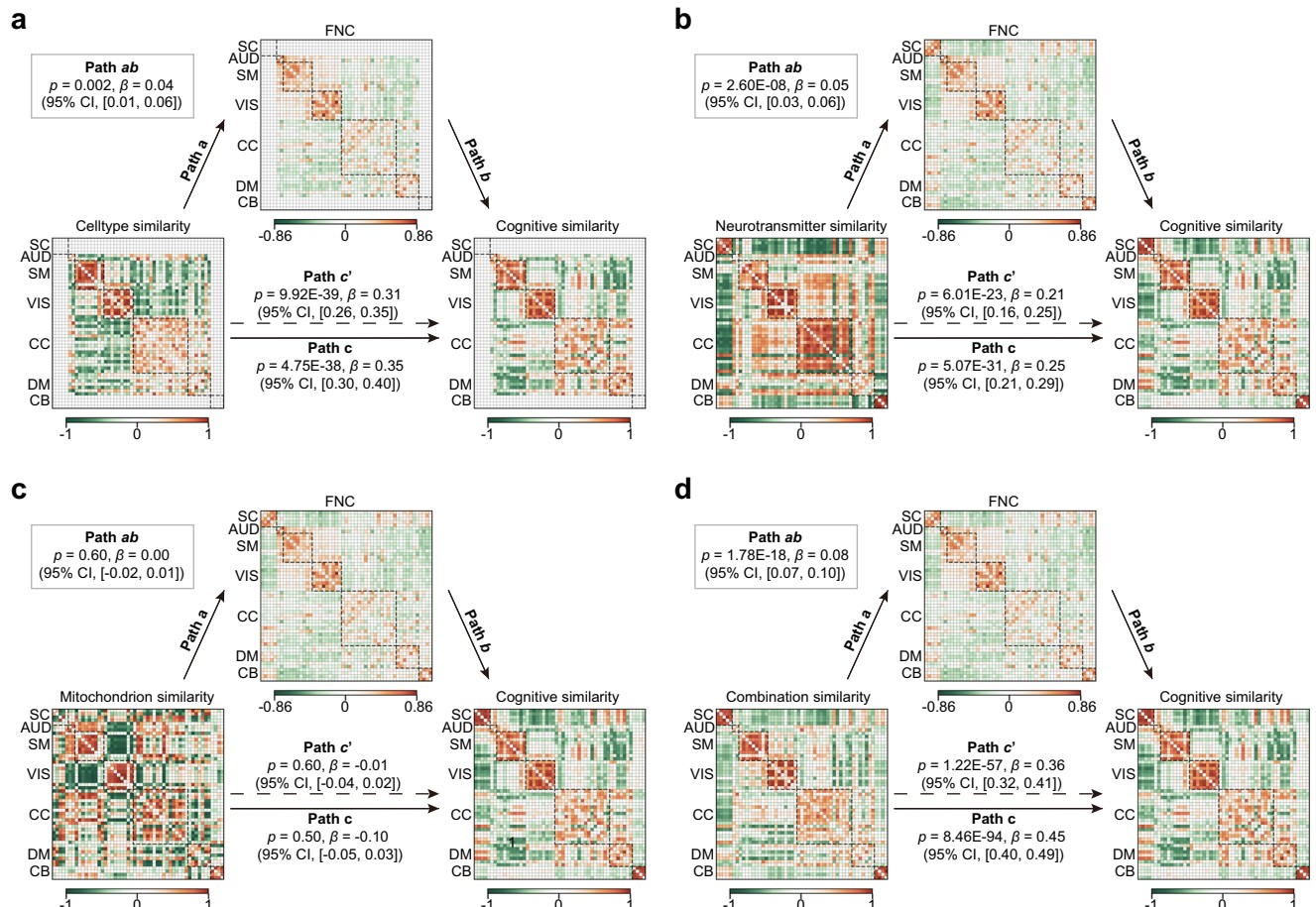

**Fig. 7 | FNC mediate the connectional associations between cellular and molecular architecture and cognitive function.** Based on 123 probabilistic cognitive activation maps derived from meta-analyses in Neurosynth[48], a cognitive similarity network was constructed by computing pairwise Spearman correlations between the spatial fingerprints of all ICN pairs. Mediation analyses at the network level treated cellular/molecular similarity as the predictor, FNC as the mediator, and cognitive similarity as the outcome. To account for spatial confounds, Euclidean distance and Dice similarity were included as covariates in mediation analyses. The 95% confidence intervals were estimated using a bias-corrected two-sided bootstrap with 1000 resamples. Significant mediation effects were observed for cell-type (**a**), neurotransmitter (**b**), and combination (**c**) similarity networks, whereas the mitochondrion similarity network (**d**) did not yield a significant mediation path. Source data are provided as a Source Data file.

interneurons in layers 2/3 and 5 elaborate axons into layer 1 and thereby preferentially suppress inputs arriving through layer 1, i.e., gate top-down signals at apical dendrites[67,68] consistent with anterior cingulate cortex (IC45) computations, which integrate cost–benefit and performance signals to guide the allocation of control[69]. Interestingly, the anatomical cores of the ICNs showing cellular associations lie at key waypoints along the cortex's principal unimodal-to-transmodal gradient[55]—anchored by IC14 (precentral gyrus) at the unimodal sensorimotor pole and extending toward IC28 (superior medial frontal gyrus) and IC45 (anterior cingulate) near the transmodal apex—while IC27 (insula) acts as switching hub between large-scale systems for external attention and self-referential cognition[70]. The cell types that capture ICN spatial variability likewise traverse this gradient, echoing evidence that functional gradients parallel systematic shifts in cellular composition[32]. Neurotransmitter receptor/transporter maps and mitochondrial phenotypes most strongly predicted ICN variation in SC domain, consistent with the high SC expression of dopaminergic and cholinergic markers (e.g., D1/D2[71,72], DAT[73], NET[74], VAChT[75]) and with the energetic specializations of subcortical circuits[12]. In contrast, within the cortex, neurotransmitter maps were more closely associated with ICNs situated along the non-waypoints of the principal gradient, seemingly reflecting scale-level specificity in the neurobiological determinants of ICN organization. Although speculative, cell-type maps, which index laminar architecture and compositional

differences[23,52,76,77], preferentially provided structural substrates for ICNs anchored at the waypoints of the gradient. By comparison, neurotransmitter maps, which capture modulatory sensitivity and synaptic communication[12,25,28,78,79], appeared to preferentially support the dynamic switching of networks positioned at non-waypoints along the gradient. Moreover, multi-scale models that combined neurobiological scales explained more ICN variance than any single scale, suggesting that functional networks may be shaped by the synergistic interactions among distinct cellular and molecular processes. This pattern is consistent with independent reports that neuromodulatory systems disproportionately shape patterns of brain structural development[37]. Notably, unlike prior parcellation-based spatial correlation approaches, the ICN-based framework offers a functionally segregated and spatially specific lens for cellular and molecular annotation. ICNs capture voxel-wise coactivation patterns associated with independent functional processes, enabling a finer-grained and functionally specific mapping between brain function and neurobiology. This approach advances beyond prior transcriptomic–functional studies by allowing the annotation of individual functional components, rather than broad cortical territories, with targeted cellular and molecular features.

Extending these findings, we observed that ICNs assigned to the same functional domain exhibited significantly greater similarity in their cellular and molecular profiles—across cellular, neurotransmitter,

and mitochondrial modalities—compared to ICNs from different domains. These similarities were paralleled by stronger functional co-activation, suggesting that shared microscale composition may facilitate coordinated activity between regions. This supports and expands upon prior work linking transcriptional profiles[29–31,80] and neuro-transmitter systems[25] to macroscale brain networks. Clustering similarity networks derived from cellular and molecular fingerprints revealed a non-random alignment with canonical functional domain assignments. Specifically, ICNs were grouped in a manner that significantly recapitulated known functional subdivisions, suggesting that the brain's cellular and molecular landscape may, in part, mirror the organization of functional networks revealed by group-average fMRI data. These results reinforce the notion that domain-specific functional architectures are supported by cell-type and neurotransmitter-specific features. Notably, this molecular-functional alignment echoes anatomical observations from histological analyzes, where homogeneous cellular composition is observed within local territories, and sharp transitions often demarcate adjacent cortical areas[81]. Interestingly, ICNs within the VIS and SM domains showed higher molecular and functional coupling than those within CC and DM domains. These domain-specific molecular architectures may also help explain observed correlation patterns of white matter networks[82] and structure–function network coupling[18,83], highlighting their role in shaping both network identity and inter-network communication. From a molecular–structural perspective, this finding supports the view that the association cortex may escape the strong structural constraints imposed by early sensory–motor cascades, thereby enabling higher-order cognitive functions beyond simple sensorimotor exchange[84].

Microscale evidence implicates cellular and molecular systems in sculpting cognitive capacities[40,85–87]. Yet, the extent to which macroscale ICNs and FNCs mediate the relationship between cellular and molecular architecture and cognition remains poorly understood. Meta-analytic cognitive atlases (e.g., Neurosynth) combine text mining, meta-analysis, and machine learning to generate probabilistic mappings between cognitive terms and neural activity patterns[48]. Building on this resource, our significant spatial association–based mediation results indicate that a set of mesoscale ICNs acts as putative interfaces through which cortical cell-type gradients relate to cognitive representations. This pattern accords with established functional accounts: the precentral gyrus network (IC14) aligns with motor circuitry and corticofugal output[88]; the insula network (IC27) supports switching between externally oriented and internally oriented/self-referential systems[70]; and the anterior cingulate cortex network (IC45) integrates cognitive control with affective/motivational processes[69,89]. SC-related ICNs anchored mediation pathways linking neurotransmitter receptor/transporter markers and mitochondrial phenotypes to broad cognitive functions, underscoring the central role of SC circuits in neuromodulatory regulation and energetic support for cognition[12,90–92]. These exploratory findings extend prior work by anchoring these functional profiles in plausible cellular architecture. From a network connectivity perspective, we observed significant associations among cellular/molecular similarity networks, static and dynamic FNCs, and the cognitive similarity network. These findings suggest that FNC—particularly its dynamic aspects—may act as a critical bridge linking microscale biological architecture to macroscale cognitive function. The observed mediation effects suggest that the alignment between brain biology and behavior is not merely correlative but may be systematically organized through the brain's dynamic communication architecture. In particular, the mediating role of dynamic FNC highlights how transient, state-dependent connectivity patterns may encode biologically meaningful variations relevant to higher-order cognition. Together, these findings move beyond simple correlations between cell types and networks or networks and cognition. Instead, they delineate a mediational pathway wherein macroscale functional networks serve as mechanistic conduits linking molecular-scale organization to behavioral phenotypes. This integrative model not only strengthens the biological plausibility of structure–function–cognition relationships but also provides a principled framework for understanding the neurobiological substrates of individual cognitive differences.

While our findings offer insights into the cellular and molecular foundations of ICNs, several limitations should be acknowledged. In this study, we derived imputed spatial cellular maps by cross-platform integration of snDrop-seq with AHBA microarray data and evaluated two deconvolution algorithms (CIBERSORTx[93] and BayesPrism[94]), which showed overall moderate concordance (Supplementary Fig. S13). Given the early stage of this field, we adopted the CIBERSORTx-derived estimates as our primary analytic input for continuity with prior work[26,32], and because a recent human-brain benchmarking study reported comparatively higher accuracy for this method[95]. Nevertheless, algorithm-dependent biases highlight the need for further data generation and head-to-head methodological evaluations tailored to this question. Second, the cognitive density maps used here are meta-analytic, inheriting heterogeneity from the underlying literature and potentially underrepresenting the multi-dimensionality of cognitive representations. Accordingly, our associational and mediation results should be interpreted as indirect, exploratory inferences rather than causal claims. Third, the current analyzes focus on healthy young adults, leaving generalizability across development, aging, and neuropsychiatric conditions unresolved; the cellular and molecular maps are donor-based, which may obscure individual-level variability related to age, sex, and other demographics; and brain-wide coverage remains uneven—particularly in subcortical and cerebellar territories and in regions with sparse receptor mapping. Looking forward, expanding spatially resolved molecular datasets (especially in these under-characterized areas) and integrating emerging individual-level multimodal resources will be essential to build more complete and personalized models of molecular–functional coupling.

In summary, this study offers a multi-scale framework linking molecular and cellular architecture to large-scale brain function by integrating transcriptomic and neurotransmitter maps with functional networks. We show that ICNs appear not only spatially aligned with, but also structurally constrained by underlying biological features—constraints that extend beyond local correspondence to reflect network-level structure. Moreover, we identify specific ICNs that mediate the translation of molecular substrates into cognitive functions, suggesting a potential bridge from microscale biology to macroscale behavior. These findings advance our understanding of how functional brain networks emerge from biological structure and laying the groundwork for integrative models that link molecular neuroscience, functional imaging, and cognition.

## Methods

### Participants and imaging preprocessing

This study complies with all relevant ethical regulations. All analyzes were performed using publicly available, de-identified human brain datasets[7,22,23,25,47,48], each of which was obtained with ethical approval in the original studies. No new human or animal subjects were involved in this work, and therefore, additional institutional ethics approval was not required. Information on participants' sex, age, consent procedures, and compensation is available in the original publications. As this study was conducted at the group and atlas levels, sex- or gender-specific analyzes were not applicable.

A total of 823 healthy young adults were selected from HCP (mean age = 28.79 ± 3.68 years; 356 males) (http://www.humanconnectomeproject.org/data) dataset[47]. Comprehensive demographic information and imaging acquisition protocols are described in reference[47]. We used preprocessed rs-fMRI data (http://www.

[humanconnectomeproject.org/data](humanconnectomeproject.org/data)). The HCP preprocessing pipeline included spatial distortion correction, motion correction, echo planar imaging distortion correction, registration to standard Montreal Neurological Institute (MNI) space, and intensity normalization. Subsequently, the data were resampled to $3 \times 3 \times 3$ mm³ isotropic voxels and spatially smoothed using a Gaussian kernel with a full width at half maximum of 6 mm. The detailed quality control procedure for all the preprocessed data can be found in the reference[7].

## ICNs
We utilized 53 ICNs from the Neuromark_fMRI_1.0[7], which are highly reproducible and biologically meaningful across independent ICA analysis of multiple large N cohorts following standard criteria for selecting ICNs. Each ICN represents a spatially independent component reflecting coherent brain activity. These ICNs were grouped into seven functional domains based on anatomical and functional priors: subcortical network (SC, 5 ICNs), auditory network (AUD, 2 ICNs), sensorimotor network (SM, 9 ICNs), visual network (VIS, 9 ICNs), cognitive-control network (CC, 17 ICNs), default-mode network (DM, 7 ICNs), and cerebellar network (CB, 4 ICNs).

## FNC
The preprocessed HCP data were decomposed into 53 subject-specific independent components and their associated TCs using a spatially constrained single-subject ICA method, with the Neuromark_fMRI_1.0 template serving as spatial priors (available via GIFT at [http://trendscenter.org/software/gift](http://trendscenter.org/software/gift))[7]. To minimize residual noise in the ICN TCs, four additional postprocessing steps were applied: (1) detrending to remove linear, quadratic, and cubic trends; (2) outlier detection and removal; (3) multiple regression of six head motion parameters (three translations and three rotations) along with their temporal derivatives; and (4) band-pass filtering (0.01–0.15 Hz). Pearson's correlations were computed between the denoised ICN time courses to generate individual-level FNC matrices, which were then averaged to obtain a group-level FNC matrix. Dynamic FNC was estimated using a sliding-window approach, segmenting the TCs into overlapping temporal windows[38]. For each segment, we computed covariance from the regularized precision matrix[96] using the graphical least absolute shrinkage and selection operator method[97] with an L1 penalty to promote sparsity. The regularization parameter ($\lambda$) was optimized for each scan by maximizing the log-likelihood on held-out data[98]. Window-wise FNC estimates were concatenated to yield a subject-level dynamic FNC representation, capturing temporal variations in ICN connectivity. To reduce redundancy, windows were subsampled along the temporal axis. Exemplar windows were selected based on local maxima in FNC variance across connections. These exemplars were clustered using k-means, repeated 500 times with random centroid initializations to ensure stability. The optimal number of states was determined via the elbow criterion[99]. The final centroids were used to assign all windows to discrete states. The fractional rate was quantified as the percentage of windows assigned to each state. For each state, dynamic FNCs were averaged across all windows assigned to that state to generate group-level state-specific FNC patterns.

## AHBA microarray data processing
Human brain gene expression data from 3702 bulk tissue samples across six postmortem brains were obtained from the AHBA dataset (http:// human.brain-map.org/)[22]. Raw data were processed at the sample level using the abagen toolbox ([https://abagen.readthedocs.io/en/stable](https://abagen.readthedocs.io/en/stable))[49,50], following a standardized and widely recommended pipeline[49]. Probe reannotation was performed based on data provided in reference[49], and probes lacking Entrez Gene IDs were excluded. To ensure reliability, only probes that exceeded background expression levels in at least 30% of all tissue samples were retained. Among these,

the probe with the highest differential stability was selected to represent each gene. After filtering and selection, a total of 16,383 genes were retained. Within each donor, gene expression values were first z-scored across genes within each sample, and then z-scored across all tissue samples.

## Cell-type deconvolution
Single-nucleus droplet-based sequencing (snDrop-seq) data[23] were obtained from the CellxGene-Census ([https://cellxgene.cziscience.com/collections/d17249d2-0e6e-4500-abb8-e6c93fa1ac6f](https://cellxgene.cziscience.com/collections/d17249d2-0e6e-4500-abb8-e6c93fa1ac6f)). Cell-type deconvolution was performed following the pipeline described previously[32]. Gene identifiers were harmonized by matching Entrez Gene IDs between the snDrop-seq and AHBA datasets, retaining only shared genes. To reduce computational load and enhance specificity, we limited the gene set to reliable marker genes consistently identified across eight cortical regions, as reported previously[23]. Transcriptionally similar cell types were grouped according to the groupings across area subclasses[23,52] to mitigate collinearity in downstream analyzes. The aggregated snDrop-seq data across all eight cortical areas were used as a reference for cell-type abundance deconvolution on each AHBA bulk-tissue sample via CIBERSORTx ([https://cibersortx.stanford.edu/](https://cibersortx.stanford.edu/))[93], a strategy also adopted in similar studies[26,32]. This resulted in estimated abundances for 24 distinct cell types across cortical samples, including L2/3IT, L4IT, L6IT, L6ITCar3, L5IT, L5ET, L5/6NP, L6CT, L6b, Chandelier, PVALB, SST, SSTCHODL, LAMP5, SNCG, PAX6, LAMP5LHX6, VIP, Astro, Endo, OPC, VLMC, Micro/PVM, and Oligo. For each AHBA donor, estimated cell-type abundances were mapped into MNI space at $3 \times 3 \times 3$ mm³ isotropic resolution using the corresponding MNI coordinates of the tissue samples. Samples mapped to the same voxel were averaged. To maximize sample retention, averaging across donors was performed only for overlapping samples, yielding a final set of 1569 samples. The resulting group-level cell-type maps were visually consistent with parcellation-level maps reported previously[32].

## Molecular atlas
We included two sets of molecular maps representing neurotransmitter systems and mitochondrial phenotypes. The neurotransmitter receptor and transporter density maps were estimated via 19 PET-derived tracer images from [https://github.com/netneurolab/hansen_receptors](https://github.com/netneurolab/hansen_receptors)[25]. The analyzed receptors and transporters included 5-HT$_{1A}$, 5-HT$_{1B}$, 5-HT$_{2A}$, 5-HT$_4$, 5-HT$_6$, 5-HTT, D$_1$, D$_2$, DAT, NET, H$_3$, $\alpha_4\beta_2$, M$_1$, VAChT, CB$_1$, MOR, NMDA, mGluR$_5$, and GABA$_A$. Six mitochondrial phenotype images were obtained from [https://neurovault.org/collections/16418/](https://neurovault.org/collections/16418/)[21]. These phenotypes included CI, CII, CIV, MitoD, TRC, and MRC. All molecular images were coregistered to MNI space with $3 \times 3 \times 3$ mm³ isotropic resolution, matching the resolution of the ICN and gene expression data to ensure consistent spatial alignment across modalities.

## Cognitive probabilistic measures
We utilized probabilistic maps from Neurosynth ([https://github.com/neurosynth/neurosynth](https://github.com/neurosynth/neurosynth))[48] to obtain voxel-wise cognitive and behavioral density maps. These maps were generated from meta-analyzes of over 15,000 published fMRI studies and quantify the probability that a given cognitive term is reported in a study, conditional on the presence of activation at a specific voxel. In total, 123 cognitive and behavioral terms were selected based on the Cognitive Atlas[56]. All cognitive association maps were resampled and aligned to MNI space with $3 \times 3 \times 3$ mm³ isotropic resolution, consistent with the spatial resolution used for functional, molecular, and gene expression data, ensuring cross-modality comparability.

## Linking ICNs with cellular and molecular atlases
To investigate the potential cellular and molecular underpinnings of individual ICNs, we assessed the spatial associations between each ICN

map and the cellular and molecular atlases. For each ICN, we applied a threshold of $|z| > 3$ (three standard deviations; a conventional choice to emphasize spatial specificity in ICA maps[100]) to exclude weakly activated voxels and to focus on the core or hub voxels that contribute most strongly to the spatially specific network pattern. Only voxels above the threshold were included in the voxel-wise spatial correlation analyzes. Because the cellular atlas is cortex-limited and discrete sampled, we implemented a local neighborhood averaging approach: for each voxel in the cellular atlas that overlapped with the thresholded ICN map, we extracted ICN values ($|z| > 3$) within a 3-mm radius and computed their mean, yielding a matched spatial representation that preserved localization while reducing noise from sampling sparsity. Accordingly, ICN-to-cell-type associations were computed only within cortical regions. Spatial similarity between ICN maps and cell-type distributions was quantified using Spearman's correlation. To account for spatial auto-correlation, we performed a two-sided Moran test[53] with 1000 permutations. For each ICN spatial map, the spatial weights matrix was defined by Euclidean distance proximity. Moran spectral randomization was then used to generate 1000 spatially autocorrelated surrogate maps that matched the observed degree of spatial dependence (Moran's $I$). Significance was determined by comparing observed statistics to null distributions, with FDR correction applied across $24 \times 44$ comparisons. For the neurotransmitter and mitochondrial atlases, we employed multivariate linear regression models[25] to estimate the contribution of each molecular map to ICN spatial variance. ICN intensity maps were treated as the outcomes, and the mitochondrial or neurotransmitter profiles were entered as predictors. The adjusted $r^2$ was used to quantify the model fit. To evaluate the relative importance of each feature, we used the relaimpo R package, which partitions the explained variance across predictors[54]. The significance of adjusted $r^2$ and relative importance was determined by one-sided Moran test[53] with 1000 permutations. The $p$-values of relative importance were corrected for ($19 \times 53$ for neurotransmitter; $6 \times 53$ for mitochondrion) comparisons using the FDR method. Robustness was assessed at alternative thresholds of 2.58 (two-sided $p = 0.01$) and 3.29 (two-sided $p = 0.001$), yielding consistent correlation results of spatial similarity across cell-type (Supplementary Fig. S14), neurotransmitter (Supplementary Fig. S15), and mitochondrion (Supplementary Fig. S16).

To investigate the synergy and relative contributions of distinct neurobiological scales to ICNs, we fitted a multi-scale model (ICN ~ all cell-type, neurotransmitter, and mitochondrion predictors) and single-scale models (one system at a time). To mitigate collinearity among cell-type proportional data while preserving interpretability, we applied factor analysis (minimum residuals, promax oblique rotation)[37]. We retained all unrotated factors that explained ≥1% of variance, and we named factors by assigning each original atlas to the factor on which it loaded most strongly. Oblique rotation was chosen because inter-correlations are expected among non-independent biological processes and cell populations. To align modalities spatially, neurotransmitter and mitochondrial maps were resampled to the discrete cortical sampling of the cell-type data using the local neighborhood averaging approach; the same sampled inputs were used when fitting the single-system models to ensure fair comparisons. Model improvement was assessed with $F$-tests comparing the multi-scale model to each single-scale model, with FDR applied across $3 \times 44 + 2 \times 9$ comparisons. Scale-level relative contributions were defined as the sum of contributions from all sub-markers within each scale.

## Associations between FNC and cellular/molecular similarity networks

We averaged cellular and molecular features across the 53 ICNs ($|z| > 3$) and constructed feature similarity networks for cell types, neuro-transmitter receptor densities, mitochondrial phenotypes, and combined molecular fingerprints using Spearman's correlation between ICNs. Robustness was assessed at alternative thresholds of 2.58 (two-

sided $p = 0.01$) and $z = 3.29$ (two-sided $p = 0.001$), yielding consistent results (Supplementary Fig. S17). To assess whether these similarity networks recapitulate functional organization, we performed a two-sided rank-sum test comparing within-domain versus between-domain connections. Importantly, to control for spatial autocorrelation, we regressed out spatial proximity (both Euclidean distance and Dice similarity) between ICN spatial maps from all connectivity matrices. Next, we assessed the global and regional alignment between the FNC and each cellular/molecular similarity network using two-sided partial Spearman's correlation, again controlling for spatial proximity. For regional correlation analyzes, multiple comparisons were controlled using FDR across the tested ICNs, 44 for cell-type analyzes and 53 for neurotransmitter, mitochondrial, and combined models.

## Structural associations between ICNs and cellular/molecular architecture

To investigate whether the functional domain structure of ICNs is underpinned by cellular and molecular organization, we first constructed cellular and molecular similarity networks while controlling for spatial autocorrelation (Euclidean distance and Dice similarity) across ICNs. We then applied a diffusion embedding method[55] to denoise and reduce the similarity network into a low-dimensional manifold representation. A three-component solution was selected via the elbow criterion, which preserved global structure while mitigating the influence of noise or local fluctuations on subsequent clustering. Effectiveness was verified by comparison with principal component analysis (PCA, first three components) and with analyzes conducted without dimensionality reduction. We then applied k-means clustering to group the projected features. The number of clusters was determined based on the number of known functional domains: 5 clusters for the cell-type similarity network and 7 clusters for the other similarity networks. The cluster labels derived from this unsupervised embedding were compared to the true functional domain labels of the ICNs. The ARI was used to quantify the accuracy of the clustering, and 1000 label permutations were performed to assess the statistical significance of the results using a one-sided permutation test.

## Mediation analysis

For cellular/molecular maps and ICNs that showed significant spatial correspondence in the primary analyzes, we then conducted mediation analyzes using the PROCESS toolbox in R. In these analyzes, the cellular/molecular maps served as the predictor, the cognitive probabilistic measures were the outcome, and the ICN acted as the mediator. The significance of the mediation was estimated by a bias-corrected two-sided bootstrap approach with 1000 random samplings. FDR was applied to correct the $p$-values for $123 \times 30$ comparisons for mediation analysis. To further investigate whether inter-ICN functional connectivity (FNC and dynamic FNC) mediates the relationship between cellular/molecular similarity and cognitive similarity, we first averaged the probabilistic values of each cognitive map across the 53 ICNs ($|z| > 3$) to obtain a single feature vector for each cognitive term. We then constructed cognitive similarity networks by computing Spearman's correlation between ICNs. Two-sided partial Spearman's correlations were performed to assess associations between cellular/molecular similarity networks and the cognitive similarity network, as well as between (static) FNC or dynamic FNC states and the cognitive similarity network. To account for spatial confounds, Euclidean distance and Dice similarity were included as covariates in all correlation and mediation analyzes. Mediation analyzes were conducted using the PROCESS, with the cellular/molecular similarity network specified as the predictor, the cognitive similarity network as the outcome, and FNC or dynamic FNC as the mediator. The 95% confidence intervals were estimated using a bias-corrected two-sided bootstrap with 1000 resamples. For the dynamic FNC models, multiple comparisons were controlled using FDR across $5 \times 4$ tests.

**Reporting summary**

Further information on research design is available in the Nature Portfolio Reporting Summary linked to this article.

## Data availability

The HCP dataset is publicly available at http://www.humanconnectomeproject.org/data. The Neuromark_fMRI_1.0 template is available at http://trendscenter.org/software/gift. The AHBA data can be accessed at http://human.brain-map.org/ and the sn-RNA dataset is available at https://cellxgene.cziscience.com/collections/d17249d2-0e6e-4500-abb8-e6c93fa1ac6f. Neurotransmitter PET imaging can be found at https://github.com/netneurolab/hansen_receptors and mitochondrial phenotype images can be found at https://neurovault.org/collections/16418/. Derived data generated in this study have been deposited at https://trendscenter.org/data/ and https://github.com/FelixFengCN/ICNs-annotation. Source data are provided with this paper.

## Code availability

SPM12 is available at http://www.fil.ion.ucl.ac.uk/spm/, and the GIFT toolbox can be accessed at http://trendscenter.org/software/gift. AHBA sample preprocessing was performed using the abagen toolbox (https://abagen.readthedocs.io/en/stable). Single-nucleus RNA-seq data processing and cell-type imputation were conducted with CIBERSORTx (https://cibersortx.stanford.edu/). Relative importance analysis was carried out using the relaimpo R package (https://cran.r-project.org/web/packages/relaimpo/index.html). Moran test and diffusion embedding analyses were implemented via BrainSpace (https://brainspace.readthedocs.io/en/latest/index.html). Mediation analyses were performed using the PROCESS R package (https://search.r-project.org/CRAN/refmans/bruceR/html/PROCESS.html). Custom analysis codes used in this study are available at https://github.com/FelixFengCN/ICNs-annotation[101].

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

## Acknowledgements

The authors thank all the volunteers for their participation in the study and anonymous reviewers for their insightful comments and suggestions. This work was supported by the National Science Foundation (NSF; Grant #2112455 to V.D.C.) and the National Institutes of Health (NIH; Grants #R01MH123610 and #R01MH136665 to V.D.C.). This work used transcriptomic data from the Allen Institute for Brain Science. Neuroimaging data were provided by the HCP, WU-Minn Consortium (principal investigators: D.V. Essen and K. Ugurbil; 1U54MH091657), funded by the 16 NIH Institutes and Centers that support the NIH Blueprint for Neuroscience Research, and by the McDonnell Center for Systems Neuroscience at Washington University. The content of this manuscript solely reflects the views of the authors and may not reflect the views of the Allen Institute, NIH or the HCP consortium investigators.

## Author contributions

G.F. and V.D.C. designed the research. G.F. conducted the research. J.C., J.S., and V.D.C. provided analytic support. G.F. wrote the manuscript and made figures. All authors edited the manuscript. V.D.C. was the project administrator.

## Competing interests

The authors declare no competing interests.
