## [Transparent Peer review file · Nature Communications]

Cellular and molecular associations with intrinsic brain organization

Corresponding Author: Professor Vince Calhoun

Version 0:

Reviewer comments:

Reviewer #1

(Remarks to the Author)

Feng et al., assess whether functionally-defined brain regions are enriched for certain cell-types, neurotransmitter receptors and mitochondrial phenotypes. The authors then show that regional similarities in cell-type and receptors markers are correlated with functional connectivity (though only to a small degree $r < 0.3$). The work builds towards a set of mediation models that suggest functional connectivity states mediate the relationship between regional similarities in microscale markers and regional similarities in cognitive affiliations. While there is definitely some interest in the field regarding how cellular and molecular architecture are related to functional differences and cognition, I did not find the present study very compelling in addressing this question.

Firstly, I struggle to see how the presently described independent component networks ("ICNs") can be considered networks. Unfortunately, the ICNs were not shown in full the present paper (this should be rectified), but upon looking at the original paper from this group on ICNs (Du et al) each component represents spatially constrained, (usually) bilateral regions. This doesn't align with the widely used definition of networks as a set of multiple interconnected regions. This has important implications for the expectations and interpretation of the present analyses.

The choice of modalities seems opportunistic rather than hypothesis-driven and has resulted in an overwhelming set of comparisons with limited synergy. The introduction provides only a very cursory overview - highlighting a small selection of previous papers in the realm of transcriptomics and neuromodulatory systems - and then ends on non-specific aims.

The discussion tends towards over-interpretation and employs causal language to describe the outcome of indirect correlations (e.g. "we demonstrate that the large-scale functional architecture of the human brain is systematically shaped by its underlying cellular and molecular organization."). For example, in the Discussion the authors state that "Each ICN exhibited a unique cell-type expression signature," when in the results only 11 of 53 ICNs show significantly unique cell-type expression.

The mediation analysis may be interesting as a preliminary suggestion of relationships between cell-types/receptors and functional states, but the link to cognition, in particular, seems dubious, given the reliance on Neurosynth maps, which represent a mixed sample of fMRI activations associated with different cognitive terms. I would expect better profiling of cognition, given the claims made.

It is not clear how FDR correction was employed, including the alpha level, accommodation for two-sided tests and across which tests the FDR was performed. This may have a major impact on significant outcomes.

Several methodological decisions are not effectively explained and seem arbitrary (e.g. threshold $|z| > 3$, use of diffusion map embedding before k-means).

Please check for typographical errors in the allocation of Figures (e.g. Figure 3D and 3E don't exist)

In the statement, "We utilized 53 ICNs from the Neuromark_fMRI_1.07, which has been shown to be meaningful", what does it mean for something to be meaningful?

More details are necessary on the implementation of the Moran permutation testing

Grammatical errors make several sentences difficult to comprehend, e.g. "Extending this analysis, we incorporated 123 cognitive and behavioral probabilistic maps from Neurosynth to assess whether functional networks (ICNs and FNCs) and mediate the relationship between microscale biological substrates and spatial and connectional variations in cognitive and behavioral functions." and "We analyzed to identify shared network associations with cellular/molecular similarity networks and FNCs."

(Remarks on code availability)

Reviewer #2

(Remarks to the Author)

Thank you for inviting me to review this manuscript by Feng and colleagues, in which the authors conduct a series of comparisons between measures taken from the brain that represent important axes of physical and structural variation (such as the genes coding for different cell types) and the spatial patterns associated with different independent components assessed from two large neuroimaging datasets. The results are compelling and paint a clear picture: there is substantial correspondence (above well-controlled chance levels) for the expression of different markers of brain organisation and the observed spread of independent components observed in human functional data.

From my perspective, I believe that the results will be of interest to the field, as there are now numerous groups interested in integrating different scales of analysis. The work aligns well with the extant literature, and while it doesn't push any conceptual boundaries with its analyses or techniques, provides a firm ground upon which the field can now ask interesting questions about different scales of brain organisation using macroscopic functional neuroimaging data.

I could not detect any obvious flaws in the authors approach, which is consistent with the known methodological rigor exhibited by this group.

The figures were excellent: clear and aesthetically-pleasing.

(Remarks on code availability)

Reviewer #3

(Remarks to the Author)

Feng and colleagues present a comprehensive analysis investigating associations between functional brain networks and cell-type gene expression, receptor density, and mitochondrial and cognitive phenotypes from both regional correspondence and interregional similarity perspectives. In general, this study provides biological explanations for ICNs by combining multi-scale high-quality datasets. However, I have several methodological concerns and questions regarding the interpretation of results that warrant clarification.

1. My primary concern is about the imputation of cell-type-specific gene expression from snDrop-seq data to AHBA bulk tissue samples. The authors use snDrop-seq data from only 8 cortical regions as a reference to impute cell-type proportions across thousands of AHBA samples spanning the entire brain. While the BayesPrism method is briefly mentioned, the validity of this critical imputation step remains inadequately described. Given that this imputed dataset forms the foundation for all subsequent analyses and conclusions in this study, it is essential to either cite and introduce previous literature demonstrating successful applications of this imputation approach to AHBA data, or present validation analyses supporting the accuracy of the imputation. Without such evidence, the reliability of all downstream results remains questionable.

2. Line 113. Regarding the voxel-based spatial correlation analysis, while this approach is appropriate for ICN maps, the methodology for mapping AHBA tissue samples to brain voxels requires clarification. Given that AHBA data consist of hundreds of discrete samples per donor rather than continuous voxel-wise measurements, how were tissue samples assigned to brain voxels? Was spatial interpolation employed? How did the authors handle those voxels without overlapping AHBA samples? These methodological details are crucial for interpreting the spatial correlation results.

3. Line 117: The selective reporting of ICN-cell-type associations lacks sufficient conclusion. What biological insights can be drawn from these associations? What hypotheses do these results support? Additionally, Figure 2A reveals that over half of the ICNs show no association with any cell type, what might explain this result? Without deeper interpretation, it's hard to draw a proper and explicit conclusion from these reported findings.

4. Similarly, the associations between ICNs and receptor density/mitochondrial phenotypes (Figure 2B) would benefit from integration within a coherent theoretical framework. Currently, the results appear as isolated findings from disparate data sources rather than components of a unified hypothesis. I strongly recommend synthesizing these findings within relevant theoretical contexts or existing literature to enhance their interpretability and impact.

5. Line 176: The demonstration that cellular/molecular similarity correlates with functional connectivity nicely confirms previous ROI-level findings. Examining the relative contributions of different ICNs or cell types to this relationship would

provide valuable mechanistic insights.

6. Figure 6: The mediation analysis framework would be more intuitive with predictors (cellular/molecular maps) positioned on the left and mediators (ICNs) in the center, following mediation diagram conventions. Additionally, the colorbar in this figure needs an interpretation.

(Remarks on code availability)

Version 1:

Reviewer comments:

Reviewer #1

(Remarks to the Author)

I found the author's response unfortunately dismissive. For example, I questioned whether the ICNs are networks because they do not involve distributed regions, but the authors simply restated how they compute these ICNs. So, generally I did not find the responses helpful. Several minor concerns have been addressed, however.

(Remarks on code availability)

Reviewer #3

(Remarks to the Author)

All my concerns have been addressed. I do not have further comment.

(Remarks on code availability)

Dear Editor and Reviewers,

We are sincerely grateful for the time and effort you devoted to evaluating our manuscript, as well as for the constructive and professional feedback you provided. Your comments have been invaluable in helping us to clarify our hypotheses, strengthen our methodological justifications, and refine our discussion of the results.

In the responses that follow, we reproduce the reviewers' original comments in *italic*, and provide our point-by-point replies underneath. Page numbers of revised sections are indicated in **bold**. Text that has been revised in the manuscript is marked in **red** in this letter and highlighted in **yellow** in the revised manuscript.

We believe that these revisions have substantially improved the clarity, rigor, and overall impact of the work. We sincerely hope that our modifications and explanations address all concerns raised.

Thank you again for your thoughtful review and dedication.

Sincerely,

Guozheng Feng and Vince D. Calhoun, on behalf of the authors

Reviewer #1 (Remarks to the Author):

Feng et al., assess whether functionally-defined brain regions are enriched for certain cell-types, neurotransmitter receptors and mitochondrial phenotypes. The authors then show that regional similarities in cell-type and receptors markers are correlated with functional connectivity (though only to a small degree $r < 0.3$). The work builds towards a set of mediation models that suggest functional connectivity states mediate the relationship between regional similarities in microscale markers and regional similarities in cognitive affiliations. While there is definitely some interest in the field regarding how cellular and molecular architecture are related to functional differences and cognition, I did not find the present study very compelling in addressing this question.

R: We sincerely thank the reviewer for their careful evaluation of our work. We greatly appreciate the recognition of the field's interest in understanding how cellular and molecular architecture related to functional differences and cognition. In the following point-by-point responses, we have clarified our hypotheses, strengthened the rationale behind our methodological choices, and provided additional analyses and explanations that we hope more compellingly demonstrate the contribution of our study.

1. Firstly, I struggle to see how the presently described independent component networks (“ICNs”) can be considered networks. Unfortunately, the ICNs were not shown in full the present paper (this should be rectified), but upon looking at the original paper from this group on ICNs (Du et al) each component represents spatially constrained, (usually) bilateral regions. This doesn't align with the widely used definition of networks as a set of multiple interconnected regions. This has important implications for the expectations and interpretation of the present analyses.

R: We thank the reviewer for this important clarification on terminology and presentation. In our manuscript, ICNs (which we define as intrinsic connectivity networks, referring to the components) denote ICA-derived resting-state brain networks. Well established since the paper¹, resting-state brain networks capture within-component coherence of spontaneous blood-oxygenation-level-dependent (BOLD) activity. Particularly, ICNs are not anatomical or parcellation-based brain atlases with predefined nodes and explicit region-to-region edges; rather, each ICN represents a spatial weight map indicating each voxel's contribution relative to the component's time course (TC). Functional network connectivity (FNC) refers to between-component connectivity, quantified as synchronization (e.g., correlations) between ICN TCs. We discuss these and other definitions of networks in an earlier paper², and the relationship between within network connectivity (ICNs) and between network connectivity (FNC) is also described in this paper³.

Importantly, our analyses focus on two ICA-based functional-network representations: (i) the spatial topology of ICN maps and (ii) the connectivity pattern captured by the FNC. We further clarified the representation of functional networks obtained via ICA-based analysis (ICNs and FNCs).

In addition, following the reviewer's suggestion, we have now included the full spatial maps of all ICNs in the revised manuscript (Supplementary Fig. S1) and cited them in the main text for transparency.

Introduction, Page 3, lines 39–45: “Among various analytic approaches, decomposition-based independent component analysis (ICA) enables the reliable extraction of intrinsic connectivity networks (ICNs), which reflect coherent patterns of spontaneous brain activity within distinct

functions⁷⁻⁹. Each ICN comprises a weight map paired with an associated time course (TC), capturing within-component connectivity in the blood oxygenation level-dependent (BOLD) signal¹⁰. Correlating these TCs yields between-component relationships, typically summarized as functional network connectivity (FNC)¹¹.”

Discussion, Page 12, lines 306–310: “Critically, unlike parcellation-based approaches, ICA enables the delineation of distinct patterns of functional activity, thereby yielding ICN maps⁵⁸. Each ICN comprises brain voxels that co-vary over time, capturing within-component connectivity. Functional connectivity between ICNs (FNC) can subsequently be derived by quantifying temporal correlations across the ICA time series.”

2. The choice of modalities seems opportunistic rather than hypothesis-driven and has resulted in an overwhelming set of comparisons with limited synergy. The introduction provides only a very cursory overview - highlighting a small selection of previous papers in the realm of transcriptomics and neuromodulatory systems - and then ends on non-specific aims.

R: We thank the reviewer for this thoughtful suggestion. Our initial hypothesis was to investigate the potential neurobiological bases of the spatial and connectional architecture of ICNs from a multi-scale perspective. The evidence we presented in the Introduction regarding associations between different biological scales and functional networks was intended to support this hypothesis. In response, we have made several major revisions to clarify the rationale for our choice of modalities, strengthen the evidence base, and clarify more specific aims.

First, we expanded the Introduction to better articulate the hypothesis-driven rationale for multi-scale perspective by integrating cell-type, neurotransmitter, and mitochondrial modalities.

Introduction, Page 3, lines 45–49: “The macroscopic functional brain networks emerge from the dynamic interplay of distributed neural circuit architectures, whose region-specific implementations—distinct cellular compositions and specialized input–output motifs—necessitate tailored neuromodulatory regulation and mitochondrial bioenergetics to sustain network-specific information processing and integration¹²⁻²¹.”

Second, we enriched the literature review to provide stronger support for scale-specific associations with functional networks. **Introduction, Page 3–4, lines 56–67:** “Previous work established spatial associations between the distribution of interneuron-linked genes and regional differences in fMRI signal variability^{26,27}. The low-dimensional manifold organizing fMRI activity shows opposing associations with regional densities of facilitatory versus inhibitory neuromodulatory receptors²⁸. In parallel, specific mitochondrial phenotypes selectively align with distinct fMRI signal metrics (e.g., maximum BOLD, regional homogeneity, entropy)²¹. Beyond spatial alignment, molecular similarity further supports the organization of functional networks: functionally coupled regions tend to share gene-expression signatures²⁹⁻³¹, and distinct cell-type distributions differentially align with the cortex’s principal unimodal-to-transmodal functional gradient³². Concordantly, neurotransmitter receptor and transporter similarity across regions correlates with functional connectivity, with brain areas exhibiting similar receptor profiles showing stronger co-activation²⁵.”

Building on prior reports, we have added the perspective supporting multi-scale synergy to the **Introduction, Page 4, lines 67–75:** “Although most prior studies have focused on individual neurobiological scales, observed correlations among neurobiological scales^{25,33} suggest that multi-

scale factors are likely to synergistically contribute to shape functional network phenotypes. One illustrative case is cholinergic neurotransmission primarily modulates neuronal excitability, regulates presynaptic transmitter release, and orchestrates the firing of neuronal populations³⁴⁻³⁶. Additionally, developmental patterns of cortical thickness are better captured by models that integrate the spatial distributions of several systems (e.g., cell-type composition, neurotransmitter receptor/transporter densities), with neurotransmitter markers showing relatively greater scale-level relative importance³⁷.”

Following the reviewer’s suggestion, we revised the last paragraph of the Introduction to explicitly clarify our overall hypotheses in the form of specific aims. **Introduction, Page 4–5, lines 92–103:** “Here, motivated by the above evidence, we aim to probe the cellular and molecular bases of the spatial and connectional architecture of ICA-derived ICNs from a multi-scale perspective. For this purpose, we integrate spatial maps of cell types, neuromodulatory receptor/transporter systems, and mitochondrial phenotypes. We hypothesize that ICN spatial maps colocalize with these microscale architectures—such that networks are differentially enriched for several biological features that support network-specific information processing and integration. We further hypothesize that between-region similarity network derived from these cellular and molecular features is associated with FNC, consistent with an organizing role for these similarity networks. Moreover, multimodal combinations of neurobiological features explain more ICN/FNC variance than any single system, consistent with potential synergistic effects among distinct microscale processes. Finally, we posit that ICNs and FNCs provide the macroscale substrates that statistically link microscale neurobiology to cognitive/behavioral maps.”

Following the hypothesis, we conducted additional experiments in the revision to support the hypothesis of multi-scale synergy. Specifically, we incorporated cell-type, neurotransmitter, and mitochondrial markers into a full model predicting ICN spatial maps, in order to test whether multi-scale integration provides greater explanatory power than single-scale models and to assess both synergistic and specific contributions. Detailed methods are described on **Page 21, lines 562–575**. Our results showed that fourteen ICNs exhibited significant model fits (adjusted r^2 : 0.62–0.99, $q < 0.05$; Supplementary Fig. S4A). In 13 of these 14 ICNs, multi-scale models outperformed single-scale models ($q < 0.05$; Supplementary Fig. S4B), underscoring the synergistic value of integrating distinct neurobiological scales. We further observed a consistent trend of synergistic contributions from markers across scales ($p < 0.05$, Supplementary Fig. S4A). Among these, neurotransmitter systems consistently accounted for the largest overall relative contributions across scales (Supplementary Fig. S4B), which aligns with previous findings⁴. These findings are reported in the Results (**Page 7–8, lines 181–189**) and further discussed in the Discussion (**Page 13–14, lines 352–356**).

3. The discussion tends towards over-interpretation and employs causal language to describe the outcome of indirect correlations (e.g. “we demonstrate that the large-scale functional architecture of the human brain is systematically shaped by its underlying cellular and molecular organization.”). For example, in the Discussion the authors state that “Each ICN exhibited a unique cell-type expression signature,” when in the results only 11 of 53 ICNs show significantly unique cell-type expression.

R: We thank the reviewer for this important observation. We agree that the correlational and

mediation analyses presented in this study reflect indirect associations and should not be interpreted in causal terms. In light of this, we carefully revised the Discussion to avoid over-interpretation and to remove causal language. Specifically, the sections referenced by the reviewer were rephrased:

Page 11, lines 292–294: “we observed spatial associations between the brain’s large-scale functional architecture and its underlying cellular and molecular organization.”

Page 12, lines 324–325: “We observe spatial coupling between the distributions of select cell types and ICNs...”

In addition, we systematically reviewed the Discussion and broadly revised phrasing wherever causal or over-interpreted statements occurred.

4. The mediation analysis may be interesting as a preliminary suggestion of relationships between cell-types/receptors and functional states, but the link to cognition, in particular, seems dubious, given the reliance on Neurosynth maps, which represent a mixed sample of fMRI activations associated with different cognitive terms. I would expect better profiling of cognition, given the claims made.

R: We thank the reviewer for this insightful comment. We understand and agree with the concern that the cognitive relevance maps derived from Neurosynth are based on a mixed meta-analytic sample of fMRI studies, which introduces heterogeneity and may underrepresent the multidimensionality of cognitive processes. Nevertheless, Neurosynth⁵ remains a widely used and valuable resource for providing large-scale, term-specific cognitive association maps. For example, prior studies have employed Neurosynth meta-analyses to reveal potential cognitive associations along the principal functional gradient⁶, to link spatial patterns of gene expression with psychological processes⁷, and to map neurotransmitter receptor distributions to 123 cognitive terms from Cognitive Atlas^{8,9}. These precedents suggest that, despite its limitations, Neurosynth offers a reasonable basis for exploratory profiling of cognition.

In our study, we position the micro–macro–cognition mediation analyses as exploratory and large-scale in scope, aimed at generating hypotheses that can inform future validation using more targeted datasets (e.g., task fMRI or individual-level cognitive measures). Consistent with the reviewer’s recommendation and our revisions in Comment 3, we have taken care to describe and interpret these results conservatively, and we explicitly acknowledged the methodological limitation in the revised manuscript in **Page 16, lines 423–427:** “Second, the cognitive density maps used here are meta-analytic, inheriting heterogeneity from the underlying literature and potentially underrepresenting the multidimensionality of cognitive representations. Accordingly, our associational and mediation results should be interpreted as indirect, exploratory inferences rather than causal claims.”

5. It is not clear how FDR correction was employed, including the alpha level, accommodation for two-sided tests and across which tests the FDR was performed. This may have a major impact on significant outcomes.

R: We thank the reviewer for this helpful suggestion. We have revised the manuscript to clarify the statistical procedures in detail, including the alpha level, the use of two-sided tests, and the scope across which FDR correction was applied.

6. Several methodological decisions are not effectively explained and seem arbitrary (e.g. threshold

$|z| > 3$, use of diffusion map embedding before k-means).

R: We thank the reviewer for this helpful suggestion. We have revised the manuscript to provide clear justifications for these methodological decisions:

Page 20, lines 535–537: “For each ICN, we applied a threshold of $|z| > 3$ (three standard deviations; a conventional choice to emphasize spatial specificity in ICA maps¹⁰⁰) to exclude weakly activated regions and to focus on spatially specific patterns.”

Page 22, lines 593–597: “We then applied a diffusion embedding method⁵⁵ to denoise and reduce the similarity network into a low-dimensional manifold representation. A three-component solution was selected via the elbow criterion, which preserved global structure while mitigating the influence of noise or local fluctuations on subsequent clustering.”, and added the cumulative variance explained by the reduced features in the Results section (**Page 9, lines 218–221**).

To assess the robustness of our findings with respect to parameter choices, we additionally tested two alternative thresholds 2.58 (two-sided $p = 0.01$) and 3.29 (two-sided $p = 0.001$) Results demonstrated high consistency across thresholds for both ICN–cellular/molecular associations (Supplementary Fig. S14–S16) and similarity networks (Supplementary Fig. S17). Although a few markers showed threshold-dependent significance, the main findings reported in this study remained robust, including key associations such as IC14–L5ET, IC27–L6ITCar3, IC28–L4IT, IC31–LAMP5, and IC45–SST.

For the choice of applying diffusion map embedding before k-means clustering, we compared performance against principal component analysis and clustering without dimensionality reduction. Diffusion embedding achieved higher scores on both the Adjusted Rand Index (ARI) and the Silhouette Score (SS), confirming its suitability for reducing similarity networks prior to clustering (Supplementary Fig. S8).

7. Please check for typographical errors in the allocation of Figures (e.g. Figure 3D and 3E don't exist)

R: We apologize for these typographical errors and thank the reviewer for the careful attention. We have thoroughly checked the manuscript to ensure that all figure references are now correct and consistent.

8. In the statement, “We utilized 53 ICNs from the Neuromark_fmri_1.0⁷, which has been shown to be meaningful”, what does it mean for something to be meaningful?

R: We appreciate the reviewer’s observation. We have revised this statement for clarity, please see **Page 17, lines 456–458:** “We utilized 53 ICNs from the Neuromark_fmri_1.0⁷, which are highly replicable and biologically meaningful ICNs across independent ICA analysis of multiple large N datasets following standard criteria for selecting ICNs.”

9. More details are necessary on the implementation of the Moran permutation testing.

R: We thank the reviewer for this helpful suggestion. We have revised the Methods to provide additional details on the implementation of Moran permutation testing, please see **Page 20, lines 544–550:** “To account for spatial autocorrelation, we performed a two-sided Moran test⁵³ with 1,000 permutations. For each ICN spatial map, the spatial weights matrix was defined by Euclidean

distance proximity. Moran spectral randomization was then used to generate 1,000 spatially autocorrelated surrogate maps that matched the observed degree of spatial dependence (Moran's *I*). Significance was determined by comparing observed statistics to null distributions, with FDR correction applied across 24×44 comparisons.”

10. *Grammatical errors make several sentences difficult to comprehend, e.g. “Extending this analysis, we incorporated 123 cognitive and behavioral probabilistic maps from Neurosynth to assess whether functional networks (ICNs and FNCs) and mediate the relationship between microscale biological substrates and spatial and connectional variations in cognitive and behavioral functions.” and “We analyzed to identify shared network associations with cellular/molecular similarity networks and FNCs.”*

R: We apologize for these grammatical errors and thank the reviewer for pointing them out. We have corrected the sentence, please see **Page 5, lines 113–116**: “**Extending this analysis, we incorporated 123 cognitive and behavioral probabilistic maps from Neurosynth⁴⁸ to assess whether functional networks (ICNs and FNCs) mediate the relationship between microscale biological substrates and spatial and connectional variations in cognitive and behavioral functions.**”

The second sentence referenced by the reviewer has been removed. In addition, we carefully reviewed the entire manuscript to ensure that similar grammatical issues do not remain.

Reviewer #2 (Remarks to the Author):

Thank you for inviting me to review this manuscript by Feng and colleagues, in which the authors conduct a series of comparisons between measures taken from the brain that represent important axes of physical and structural variation (such as the genes coding for different cell types) and the spatial patterns associated with different independent components assessed from two large neuroimaging datasets. The results are compelling and paint a clear picture: there is substantial correspondence (above well-controlled chance levels) for the expression of different markers of brain organisation and the observed spread of independent components observed in human functional data.

From my perspective, I believe that the results will be of interest to the field, as there are now numerous groups interested in integrating different scales of analysis. The work aligns well with the extant literature, and while it doesn't push any conceptual boundaries with its analyses or techniques, provides a firm ground upon which the field can now ask interesting questions about different scales of brain organisation using macroscopic functional neuroimaging data.

I could not detect any obvious flaws in the authors approach, which is consistent with the known methodological rigor exhibited by this group.

The figures were excellent: clear and asthetically-pleasing.

R: We sincerely thank the reviewer for the positive and encouraging evaluation of our work.

Reviewer #3 (Remarks to the Author):

Feng and colleagues present a comprehensive analysis investigating associations between functional brain networks and cell-type gene expression, receptor density, and mitochondrial and cognitive phenotypes from both regional correspondence and interregional similarity perspectives. In general, this study provides biological explanations for ICNs by combining multi-scale high-quality datasets. However, I have several methodological concerns and questions regarding the interpretation of results that warrant clarification.

R: We sincerely thank the reviewer for the positive evaluation of our study and for recognizing the value of integrating multi-scale datasets to provide biological explanations for ICNs. We also appreciate the reviewer's thoughtful concerns regarding methodology and interpretation. We hope that our point-by-point responses below will fully address these issues and provide the necessary clarifications.

1. My primary concern is about the imputation of cell-type-specific gene expression from snDrop-seq data to AHBA bulk tissue samples. The authors use snDrop-seq data from only 8 cortical regions as a reference to impute cell-type proportions across thousands of AHBA samples spanning the entire brain. While the BayesPrism method is briefly mentioned, the validity of this critical imputation step remains inadequately described. Given that this imputed dataset forms the foundation for all subsequent analyses and conclusions in this study, it is essential to either cite and introduce previous literature demonstrating successful applications of this imputation approach to AHBA data, or present validation analyses supporting the accuracy of the imputation. Without such evidence, the reliability of all downstream results remains questionable.

R: We sincerely thank the reviewer for this professional and insightful comment. We fully agree that cell-type deconvolution is a critical step in our study, and it is essential to evaluate its consistency with prior work.

The snDrop-seq dataset we used provides rich single-nucleus resolution, reporting 24 major cell subclasses that are consistently detected across 8 cortical regions, while exhibiting region-dependent proportion differences^{10,11}. Although this dataset does not provide high spatial resolution per se, it offers biologically robust subclass signatures that are well suited for deconvolution of cortical bulk transcriptomes. By contrast, the AHBA provides extensive coverage across thousands of samples but lacks single-cell resolution. Deconvolution is therefore the only practical strategy to estimate sample-level cell-type proportions from AHBA bulk data, and this approach has been successfully applied in previous studies linking cell-type composition to imaging phenotypes^{12,13}. For example, one recent study used the same snDrop-seq dataset to infer spatial distributions of 24 cell subclasses and demonstrated their alignment with the cortical functional gradient¹³.

Following this precedent, we decided to adopt CIBERSORTx¹⁴ as the primary method for deriving cell-type maps. Our rationale is as follows: we compared results obtained using BayesPrism¹⁵ (previously used in our analyses) and CIBERSORTx¹⁴ (used in related studies^{12,13}). We observed overall moderate concordance, but also notable differences for several cell types (Supplementary Fig. S13), likely reflecting algorithmic assumptions and differential suitability for human brain data. Importantly, a recent human-brain benchmarking study¹⁶ reported that CIBERSORTx performed more robust than BayesPrism across multiple evaluation metrics in the context of brain tissues. To

ensure continuity and comparability with prior work, we therefore re-derived our cell-type maps using CIBERSORTx (Fig. 2B), which showed strong visual consistency with parcellation-level maps previously reported¹³ (Extended Data Fig. 5–6 in the ref). We then comprehensively updated our downstream analyses based on the CIBERSORTx-derived maps:

First, we revised the Methods section to add supporting references, please see **Page 18, lines 497–498**: “Cell-type deconvolution was performed following the pipeline described previously³²”, **Page 19, lines 503–505**: “The aggregated snDrop-seq data across all eight cortical areas were used as a reference for cell-type abundance deconvolution on each AHBA bulk-tissue sample via CIBERSORTx (<https://cibersortx.stanford.edu/>)⁹³, a strategy also adopted in similar studies^{26,32}”, and **Page 19, lines 512–514**: “The resulting group-level cell-type maps were visually consistent with parcellation-level maps reported previously³².”

We also updated the results of ICN–cell type associations, please see **Page 6, lines 129–142**: “Seven ICNs exhibited at least one significant association with the imputed abundance of specific cell-types (Fig. 2A). These associations included both positive and negative correlations, indicating that cell-type gradients may support the spatial organization of ICNs. IC14 (precentral gyrus network) in the sensorimotor network (SM) domain was positively associated with L5ET ($r(107) = 0.48, q < 0.001$). IC25 (middle temporal gyrus network) in the visual network (VIS) was positively associated with L2/3IT ($r(140) = 0.30, q < 0.001$). In the cognitive-control network (CC) domain, IC27 (insula network) was significantly associated with L6ITCar3 ($r(127) = 0.58, q < 0.001$), and IC28 (superior medial frontal gyrus network) was negatively associated with L4IT ($r(123) = -0.58, q < 0.001$), IC31 (middle frontal gyrus network) was positively associated with LAMP5 ($r(110) = 0.49, q < 0.001$), IC40 (inferior frontal gyrus network) was negatively associated with L5/6NP ($r(88) = -0.40, q < 0.001$). IC45 (anterior cingulate cortex network) in the default mode domain was positively associated with SST ($r(139) = 0.32, q < 0.001$) and negative associations with OPC ($r(139) = -0.25, q < 0.001$). The spatial distributions for all 24 cell types are presented in Fig. 2B.”

The results of Cell-type and combination similarity with FNC, please see **Page 8, lines 202–203**: “... cell-type similarity (Spearman’s $r(946) = 0.19, p < 0.001$) ... and the combined similarity (Spearman’s $r(1378) = 0.46, p < 0.001$)”. Clustering analysis results, please see **Page 9, lines 226–228**: “Clustering based on cell-type (ARI = 0.34, $p < 0.001$, 5 clusters; Fig. 5A), ..., and combined profiles (ARI = 0.26, $p < 0.001$, 7 clusters; Fig. 5D) ...”

The results of ICN mediation analyses, please see **Page 10, lines 243–255**: “These analyses revealed significant mediation pathways (Extended Data Table 5), predominantly involving 7 ICNs, 10 cellular/molecular maps, and 123 cognitive and behavior functions. Among cortical ICNs (Fig. 6), the insula network (IC27; L6ITCar3-coupled) showed the broadest and strongest mediation spectrum (mediation effect β : 0.21–0.57), spanning language (speech perception/production, naming), sensory integration (multisensory, categorization), executive control (cognitive control, inhibition, decision making), and emotion-related processes (reward anticipation, anxiety, fear). The precentral gyrus network (IC14; L5ET-coupled) exhibited weaker mediation for executive/memory functions but stronger effects on perceptual attention and movement/motor control (mediation effect β : 0.10–0.23). The superior medial frontal gyrus network (IC28; L4IT-coupled) showed modest effects focused on planning and attention, with additional links to movement (mediation effect β : 0.11–0.19). Finally, the anterior cingulate cortex network (IC45; SST-coupled) preferentially

mediated affective and motivational processes, including fear, anxiety, mood regulation, and reinforcement learning (mediation effect β : 0.13–0.21).”

The results of FNC mediation analyses, please see **Page 11, lines 274–283**: “The results revealed that static FNC partially mediated the relationship between the cell-type similarity network and the cognitive similarity network (path ab : $p = 0.002$, $\beta = 0.04$, 95% CI = [0.01, 0.06]; path c' : $p < 0.001$, $\beta = 0.31$, 95% CI = [0.26, 0.35]; path c : $p < 0.001$, $\beta = 0.31$, 95% CI = [0.30, 0.40], Fig. 7A). Static FNC also mediated the link between the neurotransmitter similarity network and cognitive similarity network (path ab : $p < 0.001$, $\beta = 0.05$, 95% CI = [0.03, 0.06]; path c' : $p < 0.001$, $\beta = 0.21$, 95% CI = [0.16, 0.25]; path c : $p < 0.001$, $\beta = 0.25$, 95% CI = [0.21, 0.29], Fig. 7B), as well as between the combination similarity network and the cognitive similarity network (path ab : $p < 0.001$, $\beta = 0.08$, 95% CI = [0.07, 0.10]; path c' : $p < 0.001$, $\beta = 0.36$, 95% CI = [0.32, 0.41]; path c : $p < 0.001$, $\beta = 0.45$, 95% CI = [0.40, 0.49], Fig. 7D).”

Based on the updated results, we also revised the corresponding Discussion sections (**Page 12–14, lines 324–356; Page 15, lines 392–399**).

In addition, we explicitly clarified in the Discussion the concordance between different deconvolution algorithms and their applicability to human brain data, to provide guidance for future research, please see **Page 16, lines 416–423**: “we derived imputed spatial cellular maps by cross-platform integration of snDrop-seq with AHBA microarray data and evaluated two deconvolution algorithms (CIBERSORTx⁹³ and BayesPrism⁹⁴), which showed overall moderate concordance (Supplementary Fig. S13). Given the early stage of this field, we adopted the CIBERSORTx-derived estimates as our primary analytic input for continuity with prior work^{26,32}, and because a recent human-brain benchmarking study reported comparatively higher accuracy for this method⁹⁵. Nevertheless, algorithm-dependent biases highlight the need for further data generation and head-to-head methodological evaluations tailored to this question.”

We once again thank the reviewer for this important suggestion, which has greatly strengthened the methodological rigor of our study and ensured its continuity with prior research.

2. Line 113. Regarding the voxel-based spatial correlation analysis, while this approach is appropriate for ICN maps, the methodology for mapping AHBA tissue samples to brain voxels requires clarification. Given that AHBA data consist of hundreds of discrete samples per donor rather than continuous voxel-wise measurements, how were tissue samples assigned to brain voxels? Was spatial interpolation employed? How did the authors handle those voxels without overlapping AHBA samples? These methodological details are crucial for interpreting the spatial correlation results.

R: We thank the reviewer for this professional comment. For each AHBA donor, voxel-level gene expression data were obtained and combined with snDrop-seq reference data to estimate voxel-level cell-type abundances. These values were then mapped into MNI space at $3 \times 3 \times 3$ mm³ isotropic resolution using the MNI coordinates of the tissue samples. Tissue samples that fell into the same voxel were averaged. To maximize sample retention, averaging across donors was performed only for overlapping samples, resulting in a final set of 1,569 samples. Importantly, no spatial interpolation was employed in this process. In line with the reviewer’s suggestion, we clarified these methodological details in the revised manuscript, please see **Page 19, lines 510–514**: “For each

AHBA donor, estimated cell-type abundances were mapped into MNI space at $3 \times 3 \times 3$ mm³ isotropic resolution using the corresponding MNI coordinates of the tissue samples. Samples mapped to the same voxel were averaged. To maximize sample retention, averaging across donors was performed only for overlapping samples, yielding a final set of 1,569 samples.”

In associating ICNs with cell-type maps, we focused on voxels at the overlapping of thresholded ICN maps ($|z| > 3$) and the cell-type maps, please see **Page 20, lines 535–543**: “For each ICN, we applied a threshold of $|z| > 3$ (three standard deviations; a conventional choice to emphasize spatial specificity in ICA maps¹⁰⁰) to exclude weakly activated regions and to focus on spatially specific patterns. Only voxels above the threshold were included in the voxelwise spatial correlation analyses. Because the cellular atlas is cortex-limited and discrete sampled, we implemented a local neighborhood averaging approach: for each voxel in the cellular atlas that overlapped with the thresholded ICN map, we extracted ICN values ($|z| > 3$) within a 3-mm radius and computed their mean, yielding a matched spatial representation that preserved localization while reducing noise from sampling sparsity. Accordingly, ICN-to-cell-type associations were computed only within cortical regions.” We have provided the number of overlapping voxels between the cellular/molecular maps and each ICN in Extended Data Tables 1–3.

3. Line 117: The selective reporting of ICN-cell-type associations lacks sufficient conclusion. What biological insights can be drawn from these associations? What hypotheses do these results support? Additionally, Figure 2A reveals that over half of the ICNs show no association with any cell type, what might explain this result? Without deeper interpretation, it's hard to draw a proper and explicit conclusion from these reported findings.

R: We thank the reviewer for this thoughtful comment. Prior studies have shown that regional differences in fMRI signal variability are associated with the spatial distribution of interneuron-linked genes^{12,17}, and that distinct cell-type distributions align differentially with the cortex’s principal unimodal-to-transmodal functional gradient¹³. These findings suggest that intrinsic functional networks may exhibit cell-type-specific spatial colocalizations, pointing to distinct biological substrates. As ICA-derived ICNs reflect coherent patterns of spontaneous brain activity within distinct functions, our study specifically tested whether ICNs show cell-type-specific colocalizations consistent with network-specific information processing and integration. From our results (Fig. 2A), we observed that several ICNs—including IC14 (precentral gyrus network), IC27 (insula network), IC28 (superior medial frontal gyrus network), and IC45 (anterior cingulate network)—show significant associations with specific cell types. These associations may reflect structural and computational motifs supported by distinct neuronal populations. We expanded the Discussion to provide biological interpretations (**Page 12–13, lines 324–334**): “We observe spatial coupling between the distributions of select cell types and ICNs, consistent with shared structural and computational motifs. For example, L5ET neurons constitute the cortex’s principal corticofugal output, routing columnar computations to thalamus, brainstem, and spinal targets^{62,63}; their alignment with the precentral gyrus network (IC14) supports involvement in action control and motor execution circuits⁶⁴. By contrast, L6ITCar3 neurons possess extensive intracortical axon projections that enable cross-areal integration^{63,65}, providing a cellular substrate for the insula network’s (IC27) broad role as a switching hub across externally oriented and internally oriented systems⁶⁶. SST interneurons in layers 2/3 and 5 elaborate axons into layer 1 and thereby preferentially suppress inputs arriving through layer 1, i.e., gate top-down signals at apical

dendrites^{67,68} consistent with anterior cingulate cortex (IC45) computations, which integrate cost-benefit and performance signals to guide the allocation of control⁶⁹.”

Furthermore, we found that ICNs showing cell-type associations are located at key waypoints of the unimodal-to-transmodal cortical gradient—anchored at the sensorimotor pole (IC14), extending toward the transmodal apex (IC28, IC45), with the insula (IC27) acting as a switching hub between large-scale systems. This gradient-based organization may help explain why some ICNs show strong associations with cell types while others do not. We clarified this point in the Discussion (**Page 13, lines 334–341**): “Interestingly, the anatomical cores of the ICNs showing cellular associations lie at key waypoints along the cortex’s principal unimodal-to-transmodal gradient⁵⁵—anchored by IC14 (precentral gyrus) at the unimodal sensorimotor pole and extending toward IC28 (superior medial frontal gyrus) and IC45 (anterior cingulate) near the transmodal apex—while IC27 (insula) acts as switching hub between large-scale systems for external attention and self-referential cognition⁷⁰. The cell types that capture ICN spatial variability likewise traverse this gradient, echoing evidence that functional gradients parallel systematic shifts in cellular composition³².” In addition, we found that several ICNs not showing significant associations with cell types (“non-waypoints”) were instead related to neurotransmitter maps, which may reflect scale-level specificity in the neurobiological determinants of ICN organization (see response to Comment 4).

4. Similarly, the associations between ICNs and receptor density/mitochondrial phenotypes (Figure 2B) would benefit from integration within a coherent theoretical framework. Currently, the results appear as isolated findings from disparate data sources rather than components of a unified hypothesis. I strongly recommend synthesizing these findings within relevant theoretical contexts or existing literature to enhance their interpretability and impact.

R: In line with Reviewer #1’s Comment 2, we expanded the Introduction to strengthen the theoretical rationale for including multi-scales and to highlight both their synergistic and specific contributions. We also integrated the associations between ICNs and neurotransmitter receptor density/mitochondrial phenotypes into a broader interpretive framework in the Discussion, please see **Page 13, lines 341–347**: “Neurotransmitter receptor/transporter maps and mitochondrial phenotypes most strongly predicted ICN variation in SC domain, consistent with the high SC expression of dopaminergic and cholinergic markers (e.g., D1/D2^{71,72}, DAT⁷³, NET⁷⁴, VACHT⁷⁵) and with the energetic specializations of subcortical circuits¹². In contrast, within the cortex, neurotransmitter maps were more closely associated with ICNs situated along the non-waypoints of the principal gradient, seemingly reflecting scale-level specificity in the neurobiological determinants of ICN organization.”

In addition, to contrast cell-type versus neurotransmitter associations within cortical ICNs, we highlighted how these findings complement one another and contribute to a unified framework, please see **Page 13, lines 347–352**: “Although speculative, cell-type maps, which index laminar architecture and compositional differences^{23,52,76,77} preferentially provided structural substrates for ICNs anchored at the waypoints of the gradient. By comparison, neurotransmitter maps, which capture modulatory sensitivity and synaptic communication^{12,25,28,78,79}, appeared to preferentially support the dynamic switching of networks positioned at non-waypoints along the gradient.”

5. Line 176: The demonstration that cellular/molecular similarity correlates with functional connectivity nicely confirms previous ROI-level findings. Examining the relative contributions of

different ICNs or cell types to this relationship would provide valuable mechanistic insights.

R: We thank the reviewer for this positive feedback and constructive suggestion. In the manuscript, we present nodal-level associations between cellular/molecular similarity and functional connectivity (Fig. 4E). We found that correlations were predominantly positive, with the strongest associations observed in nodes located within the sensorimotor (SM) and visual (VIS) domains. We further elaborated on these findings in the Discussion (**Page 14, lines 377–384**): “**ICNs within the VIS and SM domains showed higher molecular and functional coupling than those within CC and DM domains. These domain-specific molecular architectures may also help explain observed correlation patterns of white matter networks⁸² and structure–function network coupling^{18,83}, highlighting their role in shaping both network identity and inter-network communication. From a molecular–structural perspective, this finding supports the view that the association cortex may escape the strong structural constraints imposed by early sensory–motor cascades, thereby enabling higher-order cognitive functions beyond simple sensorimotor exchange⁸⁴.**” At present, our similarity networks are constructed based on combined cellular features, which does not yet allow us to isolate similarity networks for specific cell types. However, we agree this is an important direction, and in future work we will consider developing approaches to quantify cell type–specific similarity networks and their associations with FNC.

6. Figure 6: The mediation analysis framework would be more intuitive with predictors (cellular/molecular maps) positioned on the left and mediators (ICNs) in the center, following mediation diagram conventions. Additionally, the colorbar in this figure needs an interpretation.

R: In line with the reviewer’s suggestion, we have revised the layout of Figure 6 to follow standard mediation diagram conventions, with predictors (cellular/molecular maps) placed on the left and mediators (ICNs) in the center. We also added an explanation of the colorbar to clarify its interpretation.

Reference

1. Greicius, M.D., Krasnow, B., Reiss, A.L. & Menon, V. Functional connectivity in the resting brain: a network analysis of the default mode hypothesis. *Proc Natl Acad Sci U S A* **100**, 253–258 (2003).
2. Erhardt, E.B., Allen, E.A., Damaraju, E. & Calhoun, V.D. On network derivation, classification, and visualization: a response to Habeck and Moeller. *Brain Connect* **1**, 1–19 (2011).
3. Joel, S.E., Caffo, B.S., van Zijl, P.C. & Pekar, J.J. On the relationship between seed-based and ICA-based measures of functional connectivity. *Magn Reson Med* **66**, 644–657 (2011).
4. Lotter, L.D., *et al.* Regional patterns of human cortex development correlate with underlying neurobiology. *Nat Commun* **15**, 7987 (2024).
5. Yarkoni, T., Poldrack, R.A., Nichols, T.E., Van Essen, D.C. & Wager, T.D. Large-scale automated synthesis of human functional neuroimaging data. *Nat Methods* **8**, 665–670 (2011).
6. Margulies, D.S., *et al.* Situating the default-mode network along a principal gradient of macroscale cortical organization. *Proc Natl Acad Sci U S A* **113**, 12574–12579 (2016).
7. Hansen, J.Y., *et al.* Mapping gene transcription and neurocognition across human neocortex. *Nat Hum Behav* **5**, 1240–1250 (2021).
8. Poldrack, R.A., *et al.* The cognitive atlas: toward a knowledge foundation for cognitive neuroscience. *Frontiers in neuroinformatics* **5**, 17 (2011).
9. Hansen, J.Y., *et al.* Mapping neurotransmitter systems to the structural and functional organization of the human neocortex. *Nat Neurosci* **25**, 1569–1581 (2022).
10. Jorstad, N.L., *et al.* Transcriptomic cytoarchitecture reveals principles of human neocortex organization. *Science* **382**, eadf6812 (2023).
11. Jorstad, N.L., *et al.* Comparative transcriptomics reveals human-specific cortical features. *Science* **382**, eade9516 (2023).
12. Anderson, K.M., *et al.* Transcriptional and imaging-genetic association of cortical interneurons, brain function, and schizophrenia risk. *Nat Commun* **11**, 2889 (2020).
13. Zhang, X.H., *et al.* The cell-type underpinnings of the human functional cortical connectome. *Nat Neurosci* **28**, 150–160 (2025).
14. Newman, A.M., *et al.* Determining cell type abundance and expression from bulk tissues with digital cytometry. *Nature biotechnology* **37**, 773–782 (2019).
15. Chu, T., Wang, Z., Pe'er, D. & Danko, C.G. Cell type and gene expression deconvolution with BayesPrism enables Bayesian integrative analysis across bulk and single-cell RNA sequencing in oncology. *Nat Cancer* **3**, 505–517 (2022).
16. Huuki-Myers, L.A., *et al.* Benchmark of cellular deconvolution methods using a multi-assay reference dataset from postmortem human prefrontal cortex. *bioRxiv* (2024).
17. Anderson, K.M., *et al.* Convergent molecular, cellular, and cortical

neuroimaging signatures of major depressive disorder. *Proc Natl Acad Sci U S A* **117**, 25138–25149 (2020).

Dear Editor and Reviewers,

We are deeply grateful for the time and effort you dedicated to evaluating our manuscript, as well as for the constructive and professional feedback that has greatly improved the quality of our work.

In the following pages, we reproduce the reviewers' original comments in *italic*, and provide our point-by-point replies underneath. Page numbers of revised sections are indicated in **bold**. Text that has been revised in the manuscript is marked in **red** in this letter.

We believe that these revisions have substantially enhanced the clarity, rigor, and overall impact of the study, and we sincerely hope that our responses fully address all concerns raised during the review process.

Thank you again for your thoughtful review and dedication.

Sincerely,

Guozheng Feng and Vince D. Calhoun, on behalf of the authors

Reviewer #1 (Remarks to the Author):

I found the author's response unfortunately dismissive. For example, I questioned whether the ICNs are networks because they do not involve distributed regions, but the authors simply restated how they compute these ICNs. So, generally I did not find the responses helpful. Several minor concerns have been addressed, however.

R: We sincerely apologize if our previous response appeared dismissive — this was certainly not our intention. We greatly appreciate the reviewer's continued attention to the interpretation of ICNs as networks, particularly given that thresholded ICN maps can appear spatially focal.

Each ICA-derived ICN is mathematically defined as a spatially distributed mode of coherent BOLD fluctuations across the brain. In other words, each ICN represents a full-brain spatial weight map, in which voxel-wise weights indicate the contribution of every voxel to a common time course (**Page 3, lines 35–40**). Thus, by definition, ICNs are distributed functional patterns that fulfill the criteria of a network, even if their core voxels appear localized after thresholding.

The threshold applied in our analyses ($|Z| > 3$) highlights the most reliable and strongly contributing voxels. This clarification has been added to the Methods (**Page 20, lines 535–538**):

“For each ICN, we applied a threshold of $|z| > 3$ (three standard deviations; a conventional choice to emphasize spatial specificity in ICA maps¹⁰⁰) to exclude weakly weighted voxels and to focus on the core or hub voxels that contribute most strongly to the spatially specific network pattern.”

Importantly, all downstream analyses were performed at the voxel level, rather than on region-averaged level. This voxel-wise framework preserves the distributed nature of ICNs and allows a direct mapping of cellular and molecular features onto their fine-grained spatial architecture—an advantage unique to ICA-based approaches compared with parcellation-based methods. We have clarified this point in the manuscript (**Page 13–14, lines 351–353**):

“ICNs capture voxel-wise coactivation patterns associated with independent functional processes, enabling a finer-grained and functionally specific mapping between brain function and neurobiology.”

We hope this clarification satisfactorily addresses the reviewer's concern.

Reviewer #3 (Remarks to the Author):

All my concerns have been addressed. I do not have further comment.

R: We sincerely thank the reviewer for the positive feedback and for recognizing that all concerns have been addressed.